# Microstructural Model of Magnetic and Deformation Behavior of Single Crystals and Polycrystals of Ferromagnetic Shape Memory Alloy

**Anatoli A. Rogovoy** *[ORCID] and **Olga S. Stolbova** [ORCID]

Institute of Continuous Media Mechanics of the Ural Branch of Russian Academy of Sciences,
614018 Perm, Russia
* Correspondence: rogovoy@icmm.ru

**Abstract:** In this article, a microstructural model of the Heusler alloy with the shape memory effect caused by the application of an external magnetic field is constructed. The dynamics of the magnetization process are described using the Landau–Lifshitz–Gilbert equation. For the numerical implementation of the model using the finite element method, the variational equations corresponding to the differential formulation of the magnetic problem are used. Such an approach makes it possible to reduce (weaken) the requirements for the smoothness of the sought solution. The problem of magnetization of single crystals of the $Ni_2MnGa$ alloy, which has a "herringbone"-type martensitic structure (a twinned variant of martensite), is considered. In each element of the twin, the magnetic domains with walls of a certain thickness are formed. The motion and interaction of these walls and the rotation of magnetization vector in the walls and domains under the action of the external differently directed magnetic fields are studied. These processes in the Heusler alloy are also accompanied by the detwinning process. A condition for the detwinning of a ferromagnetic shape memory alloy in a magnetic field is proposed, and the effect of the reorientation (detwinning) of martensitic variants forming a twin on the magnetization of the material and the occurrence of structural (detwinning) deformation in it are taken into account. First, the processes of magnetization and structural deformation in a single grain are considered at different angles between the anisotropy axes of twinned variants and the external magnetic field. For these cases, the magnetization curves are constructed, and the deformed states are identified. The model described such experimental facts as the detwinning process and the jump in magnetization on these curves as a result of this process. It was shown that the jump occurred at a certain magnitude of the strength of the applied external magnetic field and a certain direction of its action relative to the twinning system. Then, based on the obtained results, deformed states arising due to the detwinning process were determined for various (isotropic and texture-oriented) polycrystalline samples, and magnetization curves taking into account this process were constructed for these materials.

**Keywords:** micromagnetism; magnetic domains; variational formulation; finite element method; detwinning condition; polytwin crystals; magnetization curves; deformed state

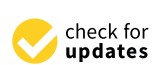

## 1. Introduction

Recently, much attention has been paid to both the practical and theoretical aspects of smart materials; that is, the materials that significantly change their physical, mechanical, or geometric characteristics under the influence of external factors of various physical natures. Such intelligent materials include the Heusler $Ni_2MnGa$ alloy, which can change its size and shape when exposed to an external magnetic field (see, for example, [1–3]). Such behavior of smart materials is associated with a phase transition of the first kind that occurs (in the absence of fields of different physical nature—force, magnetic, etc.) during the cooling (heating) of the material. When the $Ni_2MnGa$ alloy is cooled during a direct

phase transition, a cubic crystal lattice of its high-temperature phase (austenitic state) is transformed into a tetragonal crystal lattice of the low-temperature phase (martensitic state). In the above-mentioned Heusler alloy, this process is realized via a simple shift of one part of the martensitic plate relative to the other, which results in the twin formation. When a stress and/or an external magnetic field is applied to such structure, the material undergoes structural deformation of about 6–10%.

The Heusler alloys in the martensitic state at temperatures below the Curie point (376 K for $Ni_2MnGa$) are ferromagnets. Such materials exhibit spontaneous magnetization even in the absence of an external magnetic field. Each variant of martensite forming a twin has an easy magnetization axis, along which the spontaneous magnetization vector is directed. The interrelated variants of martensite, the magnetization vectors of which have the same direction, form the magnetic domains. It is energetically advantageous for the magnetic domains with differently directed magnetization vectors to be positioned in such a way that the alloy becomes non-magnetic in the absence of an external magnetic field. In a magnetic field, the walls of magnetic domains move, the magnetization vectors rotate, and the martensitic variants are reoriented (detwinned). All these processes occur at the material structure level. Therefore, in order to describe them, it is necessary to involve the microstructural modeling approach [4–7]. In contrast to the phenomenological approach, which has been the focus of many earlier publications (see, for example, [8–10]), the above-mentioned strategy allows us to construct equations that describe the process without additional assumptions.

Within the framework of the theory of micromagnetism [11], there are two approaches to describe the evolution of the magnetization vector. The first approach is based on the minimization of the magnetic energy density functional with additional restrictions on the parameters. In this case, we solve the Euler–Ostrogradsky equation, which corresponds to the minimum of this functional, or this functional is minimized directly. This approach was implemented in [12] to describe the evolution of the magnetization and to perform a numerical simulation of the motion and interaction of the Neel domain walls in a $Ni_2MnGa$ single crystal under the action of a magnetic field. The second approach uses the Landau–Lifshitz–Gilbert equation and is implemented in [13] to describe the behavior of the twinned variant of martensite, which has a more complex structure compared to the one considered in [12]. In this article, the variational equations corresponding to the Landau–Lifshitz–Gilbert differential equation and the equation for the scalar magnetic potential are written using the standard Galerkin procedure, which made it possible to reduce (weaken) the requirements for the smoothness of the solution compared to the original differential formulation (for this reason such a formulation of the problem is called weak). The capabilities of these two approaches (minimization of the magnetic energy functional and solution of the Landau–Lifshitz–Gilbert equation) to describe the magnetic processes were analyzed in work [14], and preference was given to the approach that uses the Landau–Lifshitz–Gilbert equation.

In a ferromagnetic material, purely magnetic Maxwellian stresses and mass (ponderomotive) and purely magnetic forces and moments arise due to the action of the external magnetic field. These forces and moments, which are supplements to the Maxwellian forces, cause normal elastic tension in the body. It is believed that twinning or detwinning of the martensitic structure in a ferromagnetic shape memory material occurs when the above-mentioned forces and moments reach critical values, which results in significant structural deformation (6–10%). This deformation is not magnetostrictive, which is usually neglected due to its smallness. The critical values are reached when the walls of the magnetic domains cease to move (the magnetic domains favorably located with respect to the external field grow at the expense of the domains, which have a less favorable location), and the local magnetization vectors begin to rotate in the direction preferable to the external magnetic field.

All the above processes occur in a single ferromagnetic alloy crystal with the shape memory effect. However, most of the real materials are polycrystalline materials since,

unlike single crystals, they are easier to manufacture. For example, polycrystalline films are used in spintronics, actuator, and sensor applications. However, such materials can be not only isotropic but also textured. The structures of polycrystals, which originate from single twin crystals of Ni-Mn-Ga alloys and correspond to different textures, are described in many works (see, for example, [15–19]. We will use this information while modeling the behavior of polycrystalline material based on the behavior of single crystals.

In our study, which is a continuation of our previous works [20,21], a mesostructural model of the behavior of the Heusler alloy single-twin crystal with the shape memory effect in the magnetic field has been constructed to describe such processes as the motion of magnetic domain walls, the rotation of magnetization vectors, and the reorientation (detwinning) of martensitic variants. Using this model, the magnetization curves for both the single twin crystal and polytwin crystals are constructed, and the deformation behavior of these structures is described.

The transformation of a crystal cell of one symmetry into another, the formation of twins and magnetic domains, the movement of the walls of these domains and their interaction, the rotation of magnetization vectors and the reorientation of martensite variants forming a twin, or, in other words, the detwinning process, the mechanics of the twinning and detwinning processes, and approaches to the description of magnetic and mechanical processes in these materials are those the problems of physics, mechanics, crystallography and mathematics that need to be solved to build a model that adequately describes the behavior of a ferromagnetic alloy with shape memory. Such a model is constructed and verified in this article on the basis of the results obtained earlier in the above areas by other researchers and on the authors' own results. The novelty of the approach to this study can be outlined as follows: (a) the use of the variational equations corresponding to the differential formulation of the problem of magnetization of Heusler alloys in a magnetic field, (b) the formulation of the detwinning condition for a ferromagnetic shape memory alloy under the action of the magnetic field only, (c) getting results within the framework of the approach, which is based on the variational equations and the description of the behavior of a twin martensite variant of Ni-Mn-Ga alloys in the magnetic field, taking into account the detwinning process, and (d) the construction of magnetization curves for polytwinned crystals consisting of single-twin crystals and detection of deformed states of such structures.

## 2. The Main Relations

### 2.1. The Relations of Micromagnetism

As noted in the Introduction, a direct phase transition of the first kind realized in a ferromagnetic $Ni_2Mn\,Ga$ alloy due only to its cooling leads to the fact that the martensitic plates form a twinned structure, which consists of cells having a tetragonal lattice. The boundaries of these cells may or may not coincide with the boundaries of martensitic plates, which will be discussed in more detail later. At temperatures below the Curie point, each tetragonal cell is spontaneously magnetized even in the absence of an external magnetic field. The vector of spontaneous magnetization $\boldsymbol{M}$ ($|\boldsymbol{M}| = M_s$, $M_s$ is the saturation magnetization) is directed along or against the short axis of the tetragonal lattice $c$, which is called the axis of easy magnetization. The neighboring cells, which belong to the same element as the twin and have the same directions of spontaneous magnetization vectors, form the magnetic domains. These domains are so arranged with respect to each other that the material is non-magnetized in the absence of an external magnetic field. The neighboring domains with differently oriented magnetization vectors are separated by a wall, in which the magnetization vector rotates in the direction from one domain toward the other domain. If the vector rotates in the plane of the wall, this wall is called the Bloch wall. If the vector rotates in a plane perpendicular to the plane of the wall, this wall is called the Neel wall.

In the case when the wall thickness is more than 5–10% of the domain thickness, it seems reasonable to take into account the processes occurring in the domain walls. It is

exactly this structure that is considered in our work. In the external magnetic field, the walls of these magnetic domains begin to move and interact, and the magnetization vectors in the walls and domains begin to rotate: first, the domains most favorably oriented in the direction of the applied external field increase due to less favorably oriented domains, and then the magnetization vectors in the domains try to rotate along the applied field. To describe such ferromagnetic material behavior, it is necessary to use microstructural modeling [6,11–13]. Within the framework of this simulation, the dynamics of vector $\boldsymbol{M}$ in a magnetic field are described using the Landau–Lifshitz–Gilbert equation

$$\frac{\partial \boldsymbol{m}}{\partial t} = -\gamma(\boldsymbol{m} \times \boldsymbol{H}_{eff}) + \alpha\left(\boldsymbol{m} \times \frac{\partial \boldsymbol{m}}{\partial t}\right), \tag{1}$$

where $\gamma$ is the gyromagnetic ratio, $\alpha$ is the damping (dissipation) parameter, $\boldsymbol{m} = \boldsymbol{M}/M_s$ is the unit magnetization vector, and $\boldsymbol{H}_{eff}$ is the vector of the effective field strength:

$$\boldsymbol{H}_{eff} = \boldsymbol{H}_0 - \nabla\varphi + \frac{2\,A_{exch}}{\mu_0 M_s}\,\Delta\boldsymbol{m} + \frac{2\,K_{anis}}{\mu_0 M_s}\,(\boldsymbol{m}\cdot\boldsymbol{p}^\alpha)\,\boldsymbol{p}^\alpha. \tag{2}$$

In the last expression $\boldsymbol{H}_0$ is the external field strength vector, which in our case does not depend on the coordinates (constant in space), $\mu_0$ is the magnetic constant, $A_{exch}$ is the exchange constant, $K_{anis}$ is the anisotropy constant, and $\boldsymbol{p}^\alpha$ is the direction of the easy axis of the variant $\alpha$ (in the case of the existence of several easy axes in the crystal; it is further assumed that there is only one easy axis in the crystal). $\varphi$ is the scalar depending on the vector coordinate $\boldsymbol{x}$, which determines the strength vector of the internal demagnetization field caused by the applied external field. This function satisfies the Poisson and the Laplace equations:

$$\nabla\cdot\nabla\varphi = M_s\,\nabla\cdot\boldsymbol{m} \quad \forall \boldsymbol{x} \in \Omega^{(in)}, \tag{3}$$

$$\nabla\cdot\nabla\varphi = 0 \quad \forall \boldsymbol{x} \in \Omega^{(ex)}, \tag{4}$$

where $\Omega^{(in)}$ is the region occupied by the body and $\Omega^{(ex)}$ is the region occupied by the medium surrounding the body. The function $\varphi \to 0$ for $\boldsymbol{x} \to \infty$, and on the surface $\Gamma$, which separates the body from its environment and has the external normal to the body unit vector in the actual configuration $\boldsymbol{N}$, the following equality is fulfilled:

$$2\,A_{exch}\big[\boldsymbol{m}\times(\boldsymbol{N}\cdot\nabla\boldsymbol{m})\big]\big|_\Gamma = \boldsymbol{0}, \tag{5}$$

or, taking into account that vector $(\boldsymbol{N}\cdot\nabla\boldsymbol{m})$ is perpendicular to vector $\boldsymbol{m}$ and $\boldsymbol{m} \neq \boldsymbol{0}$,

$$(\boldsymbol{N}\cdot\nabla\boldsymbol{m})\big|_\Gamma = \boldsymbol{0};$$

$$\varphi^{(in)}|_\Gamma = \varphi^{(ex)}|_\Gamma, \quad (\nabla\varphi^{(in)} - \nabla\varphi^{(ex)})|_\Gamma \cdot \boldsymbol{\mathcal{T}} = 0, \quad (\nabla\varphi^{(in)} - \nabla\varphi^{(ex)})|_\Gamma \cdot \boldsymbol{N} = M_s\,\boldsymbol{m}\cdot\boldsymbol{N}, \tag{6}$$

where $\boldsymbol{\mathcal{T}}$ is the unit tangent vector to the surface of the body $\Gamma$ in the actual configuration and the superscript $(in)$ corresponds to the body, and superscript $(ex)$ denotes its environment.

The above differential formulation of the problem requires the existence of, at least, a second derivative in the coordinates of the functions $\varphi$ and $\boldsymbol{m}$. Applying the Galerkin procedure to the Equations (1), (3) and (4) and the boundary conditions (5) and (6) described above, we have constructed the variational equations equivalent to the differential formulation of the problem (see [13,20,21]). This made it possible to reduce (weaken) the requirements for the smoothness of the sought solution (therefore, this formulation is called weak) and to use the widespread and well-proven finite element method for numerical implementation. Such a variational statement is also used in this work.

### 2.2. Twin Structure

In this subsection, we give a short explanation of the process of twin formation in order to use the constructed relations for describing the detwinning process. A detailed consideration of this process can be found in our earlier article [21]. In the phase transition of the first kind (during cooling), a cubic cell of austenite in the Ni$_2$MnGa alloy is transformed to the three tetragonal cells of martensite with pure strain tensors $\mathbf{U}_i$, $i = 1, 2, 3$ (the Bain strains), which take the following forms in the orthonormal basis $e_k$, $k = 1, 2, 3$, with vectors parallel to the edges of the cubic cell [22,23]

$$\mathbf{U}_1 = \beta\, e_1 e_1 + \alpha\, (e_2 e_2 + e_3 e_3), \qquad \mathbf{U}_2 = \alpha\, (e_1 e_1 + e_3 e_3) + \beta\, e_2 e_2,$$
$$\mathbf{U}_3 = \alpha\, (e_1 e_1 + e_2 e_2) + \beta\, e_3 e_3. \tag{7}$$

These tensors enter in the polar decompositions of the deformation gradients $\mathbf{F}_i = \mathbf{R}_i \cdot \mathbf{U}_i$ ($\mathbf{R}_i$ is the proper orthogonal tensor), which satisfy the Hadamard compatibility condition that must be met for the plane separating the two variants of martensite $i$ and $j$ with the Bain strain tensors $\mathbf{U}_i$ and $\mathbf{U}_j$ [23,24]

$$\mathbf{R}_{ij} \cdot \mathbf{U}_i = \mathbf{f} \cdot \mathbf{U}_j, \qquad \mathbf{f} = \mathbf{g} + s\, \delta_1\, \delta_2. \tag{8}$$

Here $\mathbf{f}$ is the deformation gradient describing the process of a simple shift (not to be confused with a pure shift) by the amount $s$ (by angle $\gamma$, $s = \tan \gamma$) in the plane with the unit normal $\delta_2$ in the direction of the unit vector $\delta_1$, $\mathbf{g}$ is the unit tensor and $\mathbf{R}_{ij}$ is the combination of orthogonal tensors $\mathbf{R}_i$ and $\mathbf{R}_j$: $\mathbf{R}_{ij} = \mathbf{R}_j^T \cdot \mathbf{R}_i$. The fulfillment of this condition leads to the fact that the axes of the plates of two variants of martensite with the Bain strain, for example, $\mathbf{U}_1$ and $\mathbf{U}_2$ are located at a certain angle to each other, forming a twin. Such a structure for the material under consideration can be obtained with a simple shift of a part of one of the martensite plates in the direction of the $\delta_1$ axis, as shown in Figure 1.

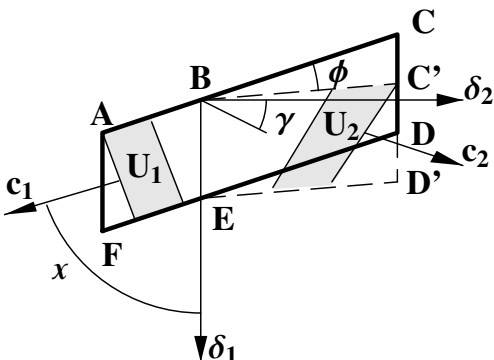

**Figure 1.** The process of twin formation with a simple shift of part of one of the martensite plate in the direction of the $\delta_1$ axis.

Here, according to the Hadamard compatibility condition, plate $ABCDEF$ with axis $c_1$ coinciding with the short axis of the tetragonal martensite cell with Bain strain $\mathbf{U}_1$ turns into a twin $ABC'D'EF$ by shifting any cross-section of part $BCDE$ of plate $ABCDEF$ parallel to the plane, to which vector $\delta_2$ is normal in the direction of vector $\delta_1$ proportional to the distance of this cross-section from the cross-section $BE$. The transformation of part $BCDE$ into part $BC'D'E$ is shown in Figure 1 by the dashed line. This corresponds to a simple shift in the direction of vector $\delta_1$ relative to vector $\delta_2$ by angle $\gamma$ or the rotation of plane $BC$ ($ED$) by angle $\phi$ to position $BC'$ ($ED'$). Angle $\chi$ is the angle between vectors $c_1$ and $\delta_1$. Again, in accordance with the Hadamard compatibility condition, the tetragonal martensite cell in plate $BC'D'E$ with Bain strain $\mathbf{U}_2$ has a short axis $c_2$, which is directed at a certain angle $\psi$ to vector $c_1$. This vector does not coincide with the axis of plate $BC'D'E$, and angle $\psi$ is not, in general, equal to $90°$. Thus, for the ferromagnetic shape memory alloy, it is necessary

to clearly differentiate between the direction of the main axes of plates forming a twin and the direction of the short axes of the tetragonal martensite cells with the Bain strains $\mathbf{U}_i$. These tetragonal martensite cells are formed in each of the two structural elements forming the twin [25], and the vectors of local spontaneous magnetization $M$ of the cells in a ferromagnetic material are directed along or against the short axes of tetragonal martensite cells $c_i$.

In Figure 1, the tetragonal martensite cells are shown as gray areas. Each tetragonal cell has two identical edges $a = b$ and one short edge, which is denoted by $c$. Therefore, the unit normal to the largest plane of the tetragonal cell is denoted by $c$ and this vector coincides with the vector of easy magnetization $p$ in the relation (2).

For Ni$_2$MnGa, the parameters of a cubic and tetragonal cell are known [26,27]: in a cubic cell, all edges are of equal length $a = 0.5852$ nm, and in a tetragonal cell, the lengths of edges are $a = b = 0.5920$ nm, $c = 0.5566$ nm, and $c/a = 0.94$. As a result, in the Bain tensors $\alpha = 0.5920/0.5852 = 1.01162$, $\beta = 0.5566/0.5852 = 0.951128$. For these parameters, it was found (see [21]) that $\chi = 46.8°$, $\gamma \approx 7°$, and $\phi \approx 4°$ and the deformation gradient, which describes a simple shift, in basis $\delta_1$, $\delta_2$ shown in Figure 1 takes the form $\mathbf{f} = \mathbf{g} + s\,\delta_1\delta_2$ (see (8)), where $s = \tan\gamma = 0.123398$, from which it follows that $\gamma \approx 7°$, in basis $e_1$, $e_2$, coinciding with vectors $c_1$, $c_2$, takes the following form

$$\mathbf{f} = \mathbf{g} - 0.061582\,(e_1 e_1 - e_2 e_2) + 0.057899\,e_1 e_2 - 0.065498\,e_2 e_1 \tag{9}$$

(regarding the relations between basis vectors $\delta_1$, $\delta_2$ and $e_1$, $e_2$ see Appendix B in [21]). For our material, the short axes $c$ of two tetragonal martensite cells, forming a twin and being also the axes of easy magnetization, are located at an angle $90°$ to each other, $\psi = 90°$, which is in full accordance with the experimental results [28–30]. (In [6], it was noted that the short axes of two tetragonal martensite cells forming a twin in the Ni$_2$MnGa alloy are at an angle of $86.5°$ to each other, and the reference was made to the experimental work of Solomon et al. [31]. However, in this publication the Ni$_{51}$Mn$_{29}$Ga$_{20}$ alloy is considered. We suppose that the difference in angles is due to this fact). Then, in basis $e_k$, for which the expressions (7) are written, vectors $c_1$ and $c_2$ in Figure 1 coincide with vectors $e_1$ and $e_2$, respectively (at $\mathbf{U}_1$ the short axis is directed along vector $e_1$ and at $\mathbf{U}_2$ along vector $e_2$), and this basis is convenient to use for describing the twin structure of the material under consideration. Axis $e_1$ of the cell triggering the twinning process (here it is a tetragonal cell with deformation $\mathbf{U}_1$) is parallel to the boundary of the twin, and axis $e_2$ of the other cell is not parallel.

## 3. Detwinning Process

As noted in the previous section, twinning in the Ni$_2$MnGa alloy occurs as a result of a simple shift. This suggests that detwinning as a reverse process should also be realized via a simple shift, which requires that certain forces must be applied on a certain surface. In the general case, these forces result from the mechanical, electrical, magnetic, and some other actions so that the detwinning process can be described in no other way than to solve the problem of the anisotropic moment theory of elasticity (see [21]). However, in the case when only a magnetic field is applied, there is an easier way to simulate the detwinning process, which is based on the calculation of the mass magnetic moment. This section is devoted to the description of such an approach.

As it follows from Figure 1, to transpose element $BC'D'E$ of twin $ABC'D'EF$ to its original position $ABCDEF$, a certain moment must be applied to it. When the external force is the magnetic field only, the mass magnetic moment $L_{mag} = \mu_0\,M \times H$, which acts in the body and in which $H = H_0 - \nabla\varphi$, is just such a moment. If the average value of vector $L_{mag}$ in the volume of element $BC'D'E$ reaches a certain critical value and is oriented in a certain direction, the detwinning process takes place and the element transforms into element $BCDE$ becoming a continuation of element $ABEF$. The disappearance of the twin structure, which complements the processes of movement and interaction of the magnetic domain walls and rotation of the magnetization vectors, takes place only in a shape memory

ferromagnetic alloy as opposed to the last two processes. To define this average critical value $L_{mag}^{cr}$, we used data from experimental works [32–35], in which the Ni-Mn-Ga alloys close to stoichiometric Ni$_2$MnGa are studied. A single-crystal prismatic sample was cut in the martensitic state from a material that experienced a direct phase transition of the first kind caused by cooling in the absence of the magnetic and force fields. The martensitic variants, which form the twin, have easy magnetization axes *c* parallel to the sides of the sample. A magnetic field was applied along one of these short axes, and the magnetization curves were plotted for these experiments. For the magnetic field $\mu_0 |H_0| = 0.3 \div 0.5$ T, these curves demonstrate a sharp jump in the magnetization. Such a jump is attributed to the reorientation of the martensitic variants that form the twin, or, in other words, to the detwinning process. Solving the magnetic problem corresponding to the experiment, it is possible to determine the average critical value of $L_{mag}^{cr}$.

Twins of the same shape can be formed in two ways. In the first case, the structure arises from element $1 - 2 - 3'$ (see Figure 2 on the left) by shifting its part $2 - 3'$ in the direction of vector $\delta_1$ to position $2 - 3$. This process was discussed in the previous subsection where, for the specific parameters of cubic and tetragonal cells for Ni$_2$MnGa, the links between the quantities present in the Hadamard compatibility equation were established.

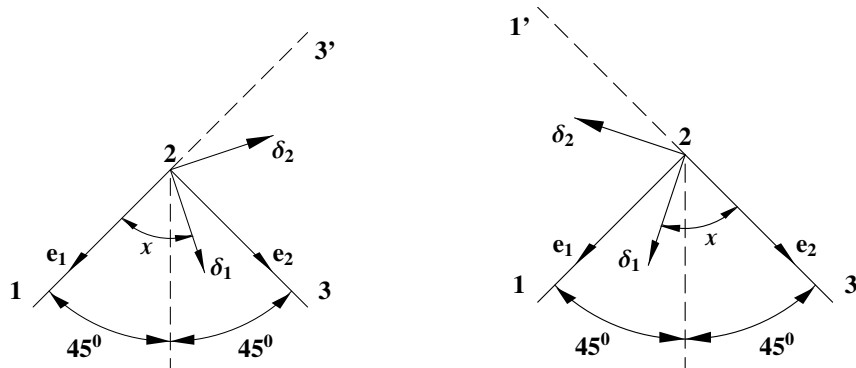

**Figure 2.** The formation of twin, $\chi = 46.8°$; twin occurs from element $1 - 2 - 3'$ by shifting its part $2 - 3'$ in direction of vector $\delta_1$ to position $2 - 3$ (on the **left**), twin occurs from element $1' - 2 - 3$ by shifting its part $1' - 2$ in the direction of vector $\delta_1$ to position $1 - 2$ (on the **right**).

As a result, the deformation gradient, describing the kinematics of the process, is constructed both for basis $\delta_i$, $i = 1, 2$, in which it is convenient to represent the deformation of a simple shear, and for basis $e_i$, $i = 1, 2$, which coincides with the axes of magnetic anisotropy of the material and also with the axes of easy magnetization. This gradient in the basis $\delta_i$ shown in Figure 2 on the left is defined, as we know, by $\mathbf{f} = \mathbf{g} + s\,\delta_1\delta_2$, where $s$ is the magnitude of the shift. As it follows from this figure,

$$\delta_1 = e_1 \cos\chi + e_2 \sin\chi, \quad \delta_2 = -e_1 \sin\chi + e_2 \cos\chi, \tag{10}$$

and then tensors $\mathbf{f}$ in basis $e_i$, which in the following will be denoted by $\mathbf{f}_1$ indicating the belonging of this tensor to the first case can be written as

$$\mathbf{f}_1 = \mathbf{g} - a\,(e_1 e_1 - e_2 e_2) + b\,e_1 e_2 - c\,e_2 e_1. \tag{11}$$

Here, for the specific values of $s$ and $\chi$ given above,

$$a = s \sin\chi \cos\chi = 0.061582, \quad b = s \cos^2\chi = 0.057899, \quad c = s \sin^2\chi = 0.065498 \tag{12}$$

and $\mathbf{f}_1$ is represented as (see (9))

$$\mathbf{f}_1 = \mathbf{g} - 0.061582\,(e_1 e_1 - e_2 e_2) + 0.057899\,e_1 e_2 - 0.065498\,e_2 e_1. \tag{13}$$

The second way of twin formation is shown in Figure 2 on the right. Here, the twin is formed from element $1' - 2 - 3$ by shifting its part $1' - 2$ in the direction of vector $\delta_1$ to position $1 - 2$. Note that vectors $\delta_1$ and $\delta_2$ on the right and left of Figure 2 have different positions relative to vectors $e_1$ and $e_2$. As follows from the solution of the Hadamard compatibility equation, the position of vector $\delta_1$, in the direction in which the shift occurs, is determined by angle $\chi$ relative to the element along which the twin is formed. In the first case, this is element $1 - 2 - 3'$, while in the second case, this is element $1' - 2 - 3$. So, the formation of twins in the second case is described by the same expression as in the first case, but for vectors $\delta_i$ shown in Figure 2 on the right, $\mathbf{f} = \mathbf{g} + s\,\delta_1\delta_2$. As it follows from this figure,

$$\delta_1 = e_1 \sin \chi + e_2 \cos \chi, \quad \delta_2 = e_1 \cos \chi - e_2 \sin \chi, \tag{14}$$

and then tensors $\mathbf{f}$ in basis $e_i$, which in the following will be denoted by $\mathbf{f}_2$ indicating the belonging to the second case takes the following form

$$\mathbf{f}_2 = \mathbf{g} + a\,(e_1e_1 - e_2e_2) - c\,e_1e_2 + b\,e_2e_1 \tag{15}$$

and with account of (12) is represented as

$$\mathbf{f}_2 = \mathbf{g} + 0.061582\,(e_1e_1 - e_2e_2) - 0.065498\,e_1e_2 + 0.057899\,e_2e_1. \tag{16}$$

In accordance with these two scenarios of twin formation, the detwinning process is also realized in two ways: when element $2 - 3$ rotates in the plane of the drawing counterclockwise relative to point 2 and takes position $2 - 3'$ and when element $1 - 2$ rotates in the plane of the drawing clockwise relative to point 2 and takes position $1' - 2$. In the first case, the positive mass magnetic moment, which is perpendicular to the plane of Figure 2 and has the required value, should be applied to element $2 - 3$, and in the second case, the negative mass magnetic moment, which is perpendicular to the plane of Figure 2 and has the required value, must be applied to element $1 - 2$. The third case, when these necessary conditions are fulfilled simultaneously for the $1 - 2$ and $2 - 3$ elements, is improbable due to all kinds of fluctuations associated with the magnetic, force, and temperature processes occurring in the body.

As mentioned above, for the external magnetic field applied along the easy magnetization axis, reorientation (detwinning) in the Heusler $Ni_2MnGa$ alloy occurs according to the experiments, when $\mu_0\,|H_0| = 0.3 \div 0.5$ T. The calculations presented in [20] show that at this point, the 180-degree walls that separate the magnetic domains disappear, but the strength of the external magnetic field is still insufficient to rotate vectors of local magnetization in the elements of the twin, mainly along the field. This situation produces a mass magnetic moment $L_{mag} = \mu_0\,M \times H$, which acts on the elements of twin $1 - 2$ and/or $2 - 3$ in Figure 2 and is mainly perpendicular to the plane of this figure. The critical value of the external field, at which the detwinning process begins, is brought into accordance with this moment.

To realize the above scenarios, a magnetic problem is solved for the twinned state of the $Ni_2MnGa$ Heusler alloy when the external magnetic field is applied along or against element $1 - 2$ (easy axis $c_1$, problem $C_1$), or along or against element $2 - 3$ (easy axis $c_2$, problem $C_2$) in Figure 2, which is in exact accordance with the experiment described above. The obtained magnetization distribution allows us to determine the average value of mass magnetic moment $L_{mag}$ in the regions occupied by elements $1 - 2$ and $2 - 3$ of the twin. In problem $C_1$, due to the motion of the magnetic domain walls, magnetization vector $M$ almost coincides with the magnetic field vector $H$ in element $1 - 2$. For this reason, mass magnetic moment $L_{mag}$ in this element of the twin is of a rather small value. In the element $2 - 3$, vector $M$, due to the motion of the magnetic domain walls and rotation of this vector, occupies such a position with respect to vector $H$ that the mass magnetic moment is directed perpendicular to the plane of the drawing when viewed from the reader (the counterclockwise rotation is realized), and its value is considered critical and denoted by $L_{mag}^{cr}$ for the above critical value of the applied external magnetic field

$\mu_0 |\boldsymbol{H}_0| = 0.3 \div 0.5$ T. This situation has a simple interpretation: the positive magnetic moment acting counterclockwise on element $2-3$ of the twin (see Figure 2) causes the rotation of this element with respect to point 2, which is also counterclockwise in the plane of the drawing. As a result, element $2-3$ becomes a linear extension of element $1-2$ (the reorientation of martensite variants occurs in accordance with the first case considered above when discussing Figure 2). In problem $C_2$, due to the motion of magnetic domain walls, magnetization vector $\boldsymbol{M}$ almost coincides with magnetic field vector $\boldsymbol{H}$ in element $2-3$. For this reason, mass magnetic moment $\boldsymbol{L}_{mag}$ in this element of the twin is of a rather small value. In element $1-2$, vector $\boldsymbol{M}$, due to the motion of the magnetic domain walls and rotation of this vector, occupies such a position with respect to vector $\boldsymbol{H}$, that the mass magnetic moment is directed perpendicular to the plane of the drawing when viewed from the reader and rotates in a clockwise direction when $|\boldsymbol{L}|_{mag} = |\boldsymbol{L}|^{cr}_{mag}$. This situation has a simple interpretation: the negative magnetic moment acting in a clockwise direction on element $1-2$ of the twin (see Figure 2) causes the rotation of this element with respect to point 2, which is also clockwise in the plane of the drawing. As a result, element $1-2$ becomes a linear extension of element $2-3$ (the reorientation of the martensitic variants occurs in accordance with the second case considered above when discussing Figure 2).

In the next section, following any of these algorithms, we will find a specific value of $|\boldsymbol{L}|^{cr}_{mag}$ corresponding to the above-mentioned experiments [32–35]. This critical value is used to determine the beginning of the detwinning process in the same material, but when the external magnetic field is applied in different directions relative to the $\boldsymbol{c}$ axis of each variant of martensite forming the twin. As noted above, detwinning occurs when the module of mass magnetic moment $|\boldsymbol{L}|_{mag}$ reaches value $|\boldsymbol{L}|^{cr}_{mag}$ and is positive in element $2-3$ of the twin with the easy axis $\boldsymbol{c}_2$ or negative in element $1-2$ of the twin with the easy axis $\boldsymbol{c}_1$. In the first case, element $2-3$ occupies position $2-3'$ and is an extension of element $1-2$. In the second case, element $1-2$ occupies position $1'-2$ (see Figure 2) and is an extension of element $2-3$. In any other cases, the detwinning process does not occur.

Kinematics corresponding to the detwinning process is described by the relations inverse of (13), (16). The latter can be easily constructed based on the expression for the deformation gradient written in basis $\delta_i$, corresponding to the left-hand and right-hand part of Figure 2, $\mathbf{f} = \mathbf{g} + s\,\delta_1\delta_2$. It is easily checked that the tensor inverse to $\mathbf{f}$ is represented as $\mathbf{f} = \mathbf{g} - s\,\delta_1\delta_2$. Substituting the Expressions (10) and (14) into this form we get

$$\mathbf{f}_1^{-1} = \mathbf{g} + a\,(\boldsymbol{e}_1\boldsymbol{e}_1 - \boldsymbol{e}_2\boldsymbol{e}_2) - b\,\boldsymbol{e}_1\boldsymbol{e}_2 + c\,\boldsymbol{e}_2\boldsymbol{e}_1 \quad \text{for the first case,}$$
$$\mathbf{f}_2^{-1} = \mathbf{g} - a\,(\boldsymbol{e}_1\boldsymbol{e}_1 - \boldsymbol{e}_2\boldsymbol{e}_2) + c\,\boldsymbol{e}_1\boldsymbol{e}_2 - b\,\boldsymbol{e}_2\boldsymbol{e}_1 \quad \text{for the second case} \tag{17}$$

and taking into account (12), we obtain the following specific expressions:

$$\mathbf{f}_1^{-1} = \mathbf{g} + 0.061582\,(\boldsymbol{e}_1\boldsymbol{e}_1 - \boldsymbol{e}_2\boldsymbol{e}_2) - 0.057899\,\boldsymbol{e}_1\boldsymbol{e}_2 + 0.065498\,\boldsymbol{e}_2\boldsymbol{e}_1 \quad \text{for the first case,}$$
$$\mathbf{f}_2^{-1} = \mathbf{g} - 0.061582\,(\boldsymbol{e}_1\boldsymbol{e}_1 - \boldsymbol{e}_2\boldsymbol{e}_2) + 0.065498\,\boldsymbol{e}_1\boldsymbol{e}_2 - 0.057899\,\boldsymbol{e}_2\boldsymbol{e}_1 \quad \text{for the second case.} \tag{18}$$

Using direct multiplication, we can easily show that tensors (17) are the inverse of tensors (11) and (15) and tensors (18) are the inverse of tensors (13) and (16).

**Remark 1.** *As noted earlier, element $2-3$ of twin $1-2-3$ (see Figure 2) passes into element $2-3'$ and becomes an extension of element $1-2$ for the first case of the detwinning process. The result of this is the elongation of the material in the direction of vector $\boldsymbol{e}_1$ and its shortening in the direction of vector $\boldsymbol{e}_2$. element $1-2$ of the twin $1-2-3$ passes into element $1'-2-3$ and becomes a continuation of element $2-3$ for the second case of the detwinning process. The result of this is the elongation of the material in the direction of vector $\boldsymbol{e}_2$ and its shortening in the direction of vector $\boldsymbol{e}_1$. As a result, the deformation process becomes most evident on this basis, and the convenience of such representation will be demonstrated below.*

## 4. Statement of the Problem and Procedure for Its Numerical Implementation

### 4.1. Computational Domain and Material Parameters

Figure 3 shows the computational domain (blue square), which is duplicated along the $x$ and $y$ axes. This domain has the "herringbone" structure consisting of a twinned variant of martensite. The short axes or axes of easy magnetization, or the anisotropy axes, are represented as $c_1$ and $c_2$. The arrows show the magnetization vectors directed along or against these axes when the external magnetic field is absent. The magnetic domains inside each martensitic variant are located at an angle of 180-degrees, while the magnetic domains belonging to the two variants of martensitic plates forming the twin are located at an angle of 90-degrees to each other. We apply the external magnetic field at different angles $\phi$ and determine the magnetic and deformation behavior of a material cell representing a grain (a monotwin crystal, single crystal) in such a field. Placing the single crystals at different angles in the plane of Figure 3, and using the results obtained, we plot magnetization curves for different polycrystalline samples and describe the deformation behavior of these materials during detwinning.

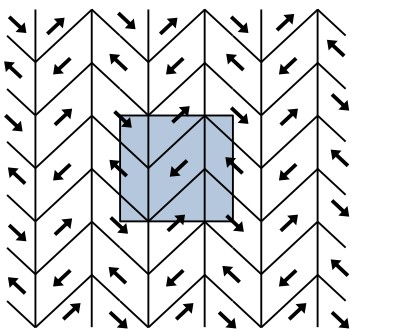 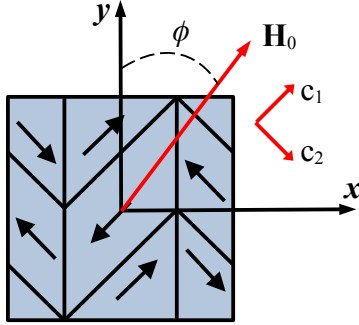

**Figure 3.** The structure of the twinned variant of martensite (on the **left**) and the computational domain in this crystal (on the **right**).

The martensitic plates in the Heusler alloy have a characteristic size of about 100–200 nm. So the size $L$ of the square computational domain $L \times L$ (blue square in Figure 3) is chosen as $L = 380$ nm. The parameters of Ni$_2$MnGa alloy are given in Table 1. For the numerical implementation of the problem, we make all the relations dimensionless by introducing the characteristic size $l_0 = 3.8$ nm and energy $\psi_0 = \mu_0 M_s^2 = 4.55 \cdot 10^5$ J/m$^3$. We estimate the thickness of the domain wall to determine the characteristic size of the finite element in the numerical calculation of the magnetization process. This size should be such that at least four to five elements fall on the domain wall. A large number of elements significantly increases the counting time; a smaller number significantly reduces the accuracy of the result. We determine the order of domain wall thickness using the relation $\Delta \approx \sqrt{A_{exch}/K_{anis}}$ given in Brown's monograph [11] and for the Ni$_2$MnGa $\Delta \approx 9$ nm. The thickness by itself, $\Delta$, is determined for a 180-degree wall in two ways: using the Lilly method $\Delta_L = \pi \delta$ and using the Landau–Lifshitz method $\Delta_{L-L} = 2\delta$, which, taking into account the above value of $\delta$, give $\Delta_L \approx 28$ nm $\approx 7.4\, l_0$ and $\Delta_{L-L} \approx 18$ nm $\approx 4.7 l_0$. As a result, we obtain the following dimensionless parameters:

$$\tilde{M}_s = 1; \quad \tilde{K}_{anis} = \frac{K_{anis}}{\psi_0} \approx 0.54; \quad \tilde{A}_{exch} = \frac{A_{exch}}{\psi_0\, l_0^2} \approx 3.$$

**Table 1.** Material Parameters

| Parameter | Value | Dimension | Source |
|:---:|:---:|:---:|:---:|
| $M_s$ | $6.015 \cdot 10^5$ | A/m | [6,36] |
| $K_{anis}$ | $2.5 \cdot 10^5$ | J/m$^3$ | [6,36] |
| $A_{exch}$ | $2 \cdot 10^{-11}$ | J/m | [6,7] |
| $\gamma$ | $2.21 \cdot 10^5$ | m/(A·s) | [6] |
| $\alpha$ | $0.5$ | - | [6] |

We also make dimensionless external magnetic field $\tilde{H}_0 = H_0/M_s$ and introduce the notation $\tilde{H}_0 = |\tilde{H}_0|$.

*4.2. Problem Formulation and Procedure for Its Numerical Implementation*

For the twinning structure, presented in Figure 3, and the shape memory ferromagnetic Ni$_2$MnGa Heusler alloy with such parameters, we formulate the problem of magnetic and deformation behavior of a monotwin and a polytwin crystal in a magnetic field.

First, the behavior of a monotwin crystal is modeled. The periodicity conditions of the solution are imposed [12]. To implement these conditions, it is necessary to consider the $2L \times 2L$ domain, in the middle of which the domain $L \times L$ (blue square in Figure 3) is located.

As noted earlier, using the initial distribution of the magnetization $m$ and the coupled variational equations, we establish the initial boundaries of the magnetic domains and determine the distribution of the magnetization vectors in them in the absence of an external magnetic field (Problem 1). The obtained magnetic structure is the initial structure to which the magnetic field is applied. Then, applying an external magnetic field in the direction of sector $c_1$ (see Figure 3) in accordance with the experiments described in the previous section [32–35], we determine the critical value of mass magnetic moment $L_{mag}^{cr}$ at which the detwinning process is realized (Problem 2). By solving this problem we will be able to describe the movement of the domain walls, the rotation of the magnetization vector, and the detwinning process that will allow us to construct the magnetization curves and define the deformation states under the action of external magnetic field $H_0$, which is applied to the computational domain shown in Figure 3 at different angles to the $y$ axis in the $(x, y)$ plane (Problem 3).

We solved the variational equation using the finite element method (FEM) using the open source computing platform FEniCS (https://fenicsproject.org, accessed on 11 January 2023). A regular grid of 5184 finite elements was used. The blue square in Figure 3 was divided into 1296 equal squares, and each of the resulting squares was divided diagonally into four equal triangles. Each element was triangle with sides 7.5 nm, 7.5 nm, and 10.6 nm. As a result, there were from three to four finite elements per domain wall, which is quite enough to ensure the necessary accuracy of the solution. The periodicity conditions of the solution, which were specified above, are imposed. A quadratic approximation was used for vector $v$, and a linear approximation for $\varphi$ and $\lambda$. $\tilde{H}_0$ increased from 0 to 1.5 with increments of $\tilde{h}_0 = 0.01$. The $\theta$-scheme was used for time stepping [13]: $m(t)$ at the current time $t$ is represented as $m_* + \theta \tau v$, where $m_* = m(t_*)$ is the magnetization at the previous time $t_*$, $\tau = t - t_*$ is the time step, $\theta \in [0, 1]$, $v = \partial m/\partial t$. The stability of the numerical solution is provided by the choice of the parameter $\theta$: if $\theta > 0.5$, the scheme will be stable for any steps in time and space. Within each step of the applied magnetic field, 3000 time steps with the values $\tau = \gamma M_s t = 0.05$ and $\theta = 0.6$ were realized to fulfill the condition of convergence of the solution.

Having a sufficient set of magnetization curves and deformed states of a single crystal at different directions of the external magnetic field application (the justification for the necessary sufficiency will be given in the following section), we describe the behavior of a polycrystal, each grain of which is a single crystal oriented in the plane of Figure 3 in a certain manner specifying the isotropic and anisotropic behavior of the polycrystal. As a result, we construct the magnetization curves and determine the deformed states

in a polycrystal, which is a representative volume of the material under consideration (Problem 4).

The results obtained by solving Problems 1–4 are given in the next section.

## 5. Results of Numerical Simulation

### 5.1. A Monotwin Crystal

As noted earlier, this work is a direct continuation of work [20]. In the last publication, the results of the calculation of the evolution of magnetization vector $m$ were presented for the material and the computational domain considered in this article. Firstly, the coupled variational equations corresponding to the differential formulation of the magnetic problem (see Section 2.1) were solved for the initial distribution of magnetization vector $m$, shown in Figure 3, in the absence of an external magnetic field. This is a solution to Problem 1 posed in Section 4.2. Such a magnetic structure is considered to be initial for the subsequent application of the external magnetic field at different angles $\phi$ to the $y$ axis (see Figure 3). In our case, $0° \leq \phi \leq 90°$ and, as will be shown below, such a change in the angle is enough to describe the magnetic and deformation behavior of various polytwin crystals. The obtained results demonstrate that at the initial stage, the magnetization occurs due to the motion of the magnetic domain walls and also due to the rotation of the magnetization vectors.

In work [20] discussed above, the detwinning process was not taken into account. However, knowing the distribution of the magnetization vector in the computational domain for angles $0° \leq \phi \leq 90°$, we can construct the dimensionless mass magnetic moments $\tilde{L}_{mag} = L_{mag}/\psi_0$ corresponding to these angles, where $L_{mag} = \mu_0 M \times H$. $\tilde{L}_{mag}$ is defined as the average value of the mass magnetic moment in elements $1-2$ (see Figure 2) corresponding to the center of the computational domain in Figure 3, for this domain, and in elements $2-3$ corresponding to the periphery of this computational domain, for this domain. The key results applicable for further explanation and use are shown in Figure 4 depending on $\tilde{H}_0$. The blue color shows the moment for element $1-2$ (the middle area in Figure 3), and red for element $2-3$ (the periphery area in Figure 3). $\phi$ is the angle to the $y$ axis in degrees.

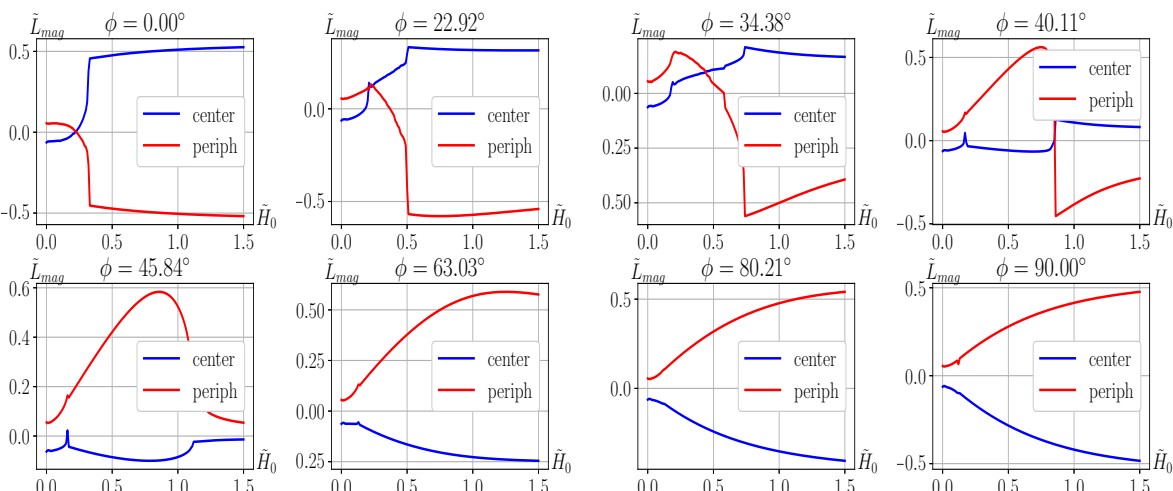

**Figure 4.** Mass magnetic moments at different angles $\phi$ to the $y$ axis, the blue color shows the moment for element $1-2$ (the middle area in Figure 3), red for element $2-3$ (the periphery area in Figure 3).

The dependence of $\tilde{L}_{mag}$ on $\tilde{H}_0$, shown in Figure 3 for angle $\phi = 45.84°$, allows us to solve Problem 2 formulated in Section 4.2 and determine the critical value of mass magnetic moment $\tilde{L}_{mag}^{cr}$, at which the detwinning process is realized. Here, the vector of external magnetic field $H_0$ acts along the axis of easy magnetization $c_1$ of one of the elements of the twin, located in the central region of Figure 3, the same as in experiments [32–35], in which the magnetization curve is constructed taking into account the detwinning process.

As follows from Figure 6 *d* of article [20], 180-degree walls separating the magnetic domains disappear at angle $\phi = 45.84°$ already for $\tilde{H}_0 = 0.17$, which corresponds to $\mu_0 |H_0| = 0.13$ T, and vector $m$ lies completely in the plane of this figure. For this reason, the vector of mass magnetic moment $\tilde{L}_{mag}$ in Figure 4 (its average value) at angle $\phi = 45.84°$ is directed perpendicular to the plane of this figure when viewed from the reader (red line, counterclockwise rotation, positive moment $\tilde{L}_{mag}$) in the peripheral region of Figure 3, where the vector of easy magnetization is $c_2$, or is also perpendicular to the plane of this figure (blue line, clockwise rotation, negative moment $\tilde{L}_{mag}$) in the middle area of Figure 3, where the vector of easy magnetization is $c_1$. The directions of these moments in these areas correspond to all detwinning cases discussed in Section 3. However, due to a very small value of $\tilde{L}_{mag}$ in the central region, detwinning will occur in accordance with the first case, when the mass magnetic moment in the peripheral area reaches a critical value $|\tilde{L}|_{mag}^{cr}$. As follows from the experiments presented in works [32–35], detwinning begins when the external magnetic field reaches value $\mu_0 |H_0| = 0.3 \div 0.5$ T. For the smallest value $\mu_0 |H_0| = 0.3$, to which $\tilde{H}_0^{cr} = 0.41$ corresponds, from Figure 4 we find that for angle $\phi = 45.84°$, $|\tilde{L}|_{mag}^{cr} = 0.35$, $|L|_{mag}^{cr} = 0.16 \cdot 10^6$ N $\cdot$m/m$^3$ and this is the solution to Problem 2 posed in subsection 4.2. The distributions of $\tilde{L}_{mag}$, including those shown in Figure 4, and the obtained critical value of the mass magnetic moment allow us to conclude that there is no detwinning until $\phi$ is less than $\approx 40°$. In the area of $(\approx 40°) \le \phi \le (\approx 89°)$, detwinning occurs in accordance with the first case. At the same time, $\tilde{H}_0^{cr}$, at which the critical value $|\tilde{L}|_{mag}^{cr}$ is reached, varies from 0.40 at $\phi \approx 40°$ to 0.72 at $\phi \approx 89°$. At the angle of 90°, the curves of the dependence of $\tilde{L}_{mag}$ on $\tilde{H}_0$ for element $1 - 2$ (the mid-area of the calculated domain in Figure 3) and for element $2 - 3$ (the periphery of the computational domain in Figure 3) differ only by a sign (see Figure 4). This means that in this case, detwinning can occur both in accordance with the first and the second case but only when the external magnetic field reaches value $\tilde{H}_0^{cr} = 0.72$. Physically, the probability of any of these cases is the same due to all kinds of fluctuations accompanying the magnetic, force, and temperature processes occurring in the medium. However, as shown below, at $\phi = 90° + \gamma$, $0° < \gamma < 90°$, detwinning occurs in accordance with the second case. Therefore, from the viewpoint of mathematics, when approaching the angle of 90° from the side 90°$-$, then detwinning occurs in accordance with the first case, but when approaching it from the side 90°$+$, detwinning occurs in accordance with the second case. Without having specific physical data, we will continue to adhere to this viewpoint.

To construct models of polytwin crystal behavior, the dependencies of $\tilde{L}_{mag}$ on $\tilde{H}_0$ were calculated for 17 values of angles $\phi$, including 8 shown in Figure 4. The segment along angle $\phi$ from 0 to 1.5 radians was passed in increments of 0.1 radians, 1.5708 radians corresponding to 90°. Table 2 shows the values of $\tilde{H}_0^{cr}$, at which the first or the second case of the detwinning process occurs for the corresponding $\phi$. As noted earlier, detwinning does not occur for angles $\phi$ less than 40°. Therefore, the table shows the results only for the angles greater than 40°.

**Table 2.** The values, at which the detwinning process occurs

| $\phi$, radians | 0.7 | 0.8 | 0.9 | 1.0 | 1.1 | 1.2 | 1.3 | 1.4 | 1.5 | 1.5708 |
|---|---|---|---|---|---|---|---|---|---|---|
| $\phi$, degrees | 40.11 | 45.84 | 51.57 | 57.30 | 63.03 | 68.75 | 74.48 | 80.21 | 85.94 | 90.00 |
| $\tilde{H}_0^{cr}$ | 0.40 | 0.41 | 0.42 | 0.43 | 0.45 | 0.48 | 0.52 | 0.57 | 0.65 | 0.72 |
| Detwinning | | | | | 1 case | | | | | 1 or 2 case |

Such a change in angle $\phi$ is sufficient to describe the magnetic and deformation reaction of the material to the application of an external magnetic field at the angles from 0° to 360° completely, provided that we take into account the following. Figure 3 shows the computational domain, which is duplicated in the horizontal and vertical directions. Such a choice of the computational domain allows us to set the periodicity condition of the solution.

However, the above choice of domain duplication in space is not the only one. In Figure 5, in addition to the computational domain, another domain containing a twin and having a certain symmetry is presented (let's denote it as $D_S$). Further analysis will be carried out based on the symmetry. In Figure 6, the magnetic structure of this symmetrical domain $D_S$ is shown. Here, the $k_2$ axis is the vertical axis $y$ of symmetry for the domain $D_S$ and axis $k_1$ is axis $x$ in Figure 5. All angles between axes $k_1$, 01′, $k_2$, 02′, $-k_1$, 01, $-k_2$, 02, $k_1$ are equal to 45°. The vectors of spontaneous magnetization $m$ are directed from point 0 to points 1 and 2′ for one group of magnetic domains and from point 0 to points 2 and 1′ for another group (see Figures 5 and 6).

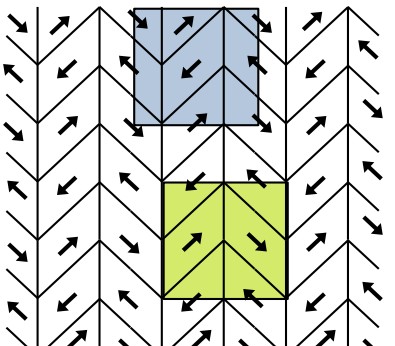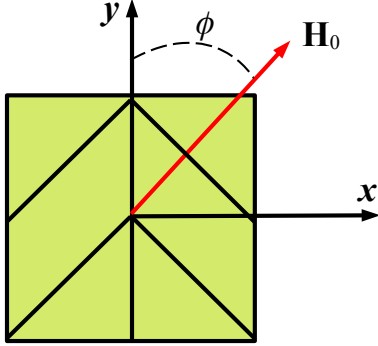

**Figure 5.** The structure of the twinned variant of martensite (on the **left**) and domain $D_S$ containing a twin and having a certain symmetry (on the **right**).

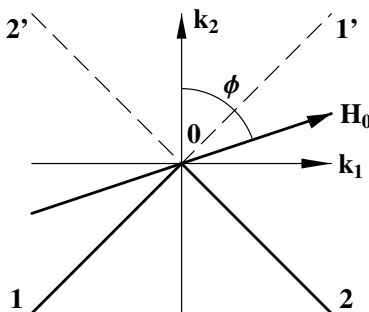

**Figure 6.** The magnetic structure of symmetrical domain $D_S$. $H_0$ is a strength vector of the external magnetic field applied to the domain at an angle $\phi$.

The magnetic structure presented in Figure 6 has four 180-degree axes of symmetry: $k_1$, $k_2$ and two diagonals $1 - 1′$ and $2 - 2′$. The symmetry with respect to diagonals replaces vectors $k_1$ and $k_2$ with vectors $k_2$ and $k_1$ or $-k_2$, and $-k_1$, respectively. The symmetry with respect to vector $k_1$ replaces vector $k_2$ with vector $-k_2$ and symmetry with respect to vector $k_2$ replaces vector $k_1$ with vector $-k_1$. We use symmetries with respect to vectors $k_1$ and $k_2$ based on the fact that mutual positions of vectors $m$ and $H_0$ correspond to the conditions of such symmetry. We describe the symmetry using an orthogonal tensor [37].

$$\mathbf{O}(\varphi, e) = \mathbf{g} \cos \varphi + ee\,(1 - \cos \varphi) + (e \times \mathbf{g}) \sin \varphi. \tag{19}$$

This tensor rotates any vector $a$ with respect to vector $e$ by an angle $\varphi$ counterclockwise, leaving its length unchanged: $a' = \mathbf{O}(\varphi, e) \cdot a$, $|a'| = |a|$. Moreover, for this transformation, the angle between the two vectors $a$ and $b$ remains unchanged: $a \cdot b = a' \cdot b'$.

Using Expression (19), a 180-degree rotation relative to the $k_1$ axis can be described by the orthogonal tensor $\mathbf{O}(k_1) = k_1 k_1 - k_2 k_2 - k_3 k_3$. As noted earlier, although the problem under consideration is a plane, when solving the Landau–Lifshitz–Gilbert equation, we deal with vector $m$ that has three components, $m = m^1 k_1 + m^2 k_2 + m^3 k_3$. However, there is

no reason to take into account $m^3$ when calculating the magnetization curves and the mass magnetic moment for which we carry out this analysis. $m^3$ does not affect the magnetization curves, as will be shown below, and the magnitude of the mass magnetic moment required to determine its critical value is calculated when $m^3$ is already zero. Therefore, in the following, the components of any vector or tensor associated with vector $k_3$ are not taken into account. Bearing in mind the above consideration, $\mathbf{O}(k_1) = k_1 k_1 - k_2 k_2$ and vectors

$$\boldsymbol{m} = m^1\,k_1 + m^2\,k_2, \quad \boldsymbol{H}_0 = H_0^1\,k_1 + H_0^2\,k_2, \quad \boldsymbol{H} = H^1\,k_1 + H^2\,k_2 \tag{20}$$

are transformed by such rotation into vectors $\boldsymbol{m}' = \mathbf{O} \cdot \boldsymbol{m} = m^1\,k_1 - m^2\,k_2$, $\boldsymbol{H}_0' = \mathbf{O} \cdot \boldsymbol{H}_0 = H_0^1\,k_1 - H_0^2\,k_2$, and $\boldsymbol{H}' = \mathbf{O} \cdot \boldsymbol{H} = H^1\,k_1 - H^2\,k_2$ (vectors $\boldsymbol{m}'$, $\boldsymbol{H}_0'$, and $\boldsymbol{H}'$ are a mirror image relative to axis $k_1$ of vectors $\boldsymbol{m}$, $\boldsymbol{H}_0$ and $\boldsymbol{H}$). The magnetization curves, which are described by the relation $\tilde{m} = \boldsymbol{m} \cdot (\boldsymbol{H}_0/H_0)$ remain unchanged in this case since $\boldsymbol{m} \cdot (\boldsymbol{H}_0/H_0) = \boldsymbol{m}' \cdot (\boldsymbol{H}_0'/H_0')$, but vector $\boldsymbol{L}_{mag} = \mu_0\,M_s\,\boldsymbol{m} \times \boldsymbol{H}$ changes its sign because $\boldsymbol{m} \times \boldsymbol{H} = -\,\boldsymbol{m}' \times \boldsymbol{H}'$ (both can be easily verified using a simple substitution. Note that $m^3$ does not affect $\boldsymbol{m} \cdot (\boldsymbol{H}_0/H_0)$ at all). Due to such rotation, angle $\phi$ between vector $\boldsymbol{H}_0'$ and the fixed vertical axis $y$ (see Figure 5), which is equal to $\beta$ for vector $\boldsymbol{H}_0$, becomes equal to $\pi - \beta$, $0 \le \beta \le \pi/2$ (mathematically, this follows from the analysis of scalar products $k_2 \cdot \boldsymbol{H}_0$ and $k_2 \cdot \boldsymbol{H}_0'$). Bearing this in mind and considering the results of Section 3, we conclude that the detwinning process for $\pi - \beta$ begins at the same $\tilde{H}_0^{cr}$ as is given in Table 2 for angle $\phi = \beta$ but unlike the first case it will be carried out in accordance with the second case discussed in Section 3. The magnetization curves will be exactly the same as for angle $\phi = \beta$.

Now, let us perform a 180-degree rotation around vector $k_2$, $\mathbf{O}(k_2) = -k_1 k_1 + k_2 k_2$, in addition to the previous rotation: $\mathbf{O} = \mathbf{O}(k_2) \cdot \mathbf{O}(k_1) = -k_1 k_1 - k_2 k_2$. It is to be noted that this product is commutative, in contrast to the general case. As a result, vectors $\boldsymbol{m}$, $\boldsymbol{H}_0$ and $\boldsymbol{H}$ (20) are transformed by such rotation into vectors $\boldsymbol{m}' = \mathbf{O} \cdot \boldsymbol{m} = -(m^1\,k_1 + m^2\,k_2)$, $\boldsymbol{H}_0' = \mathbf{O} \cdot \boldsymbol{H}_0 = -(H_0^1\,k_1 + H_0^2\,k_2)$ and $\boldsymbol{H}' = \mathbf{O} \cdot \boldsymbol{H} = -(H^1\,k_1 + H^2\,k_2)$ that leads to the equalities $\tilde{m} = \tilde{m}'$ and $\boldsymbol{L}_{mag} = \boldsymbol{L}_{mag}'$. Given that angle $\phi$ between vector $\boldsymbol{H}_0'$ and the fixed vertical axis $y$ shown in Figure 5, which is equal to $\beta$ for vector $\boldsymbol{H}_0$, becomes equal to $\pi + \beta$, $0 \le \beta \le \pi/2$, we conclude that detwinning process for $\pi + \beta$ begins at the same $\tilde{H}_0^{cr}$ as is given in Table 2 for angle $\phi = \beta$ and is carried out in accordance with the first case indicated in this table. The magnetization curves will be exactly the same as for angle $\phi = \beta$.

Finally, we will perform a 180-degree rotation only around the $k_2$ axis: $\mathbf{O}(k_2) = -k_1 k_1 + k_2 k_2$. Of course, this rotation is a 180-degree rotation around vector $k_1$ in addition to the previous rotation $\mathbf{O} = \mathbf{O}(k_2) \cdot \mathbf{O}(k_1)$. vectors $\boldsymbol{m}$, $\boldsymbol{H}_0$ and $\boldsymbol{H}$ (20) are transformed by such rotation into vectors $\boldsymbol{m}' = \mathbf{O} \cdot \boldsymbol{m} = -m^1\,k_1 + m^2\,k_2$, $\boldsymbol{H}_0' = \mathbf{O} \cdot \boldsymbol{H}_0 = -H_0^1\,k_1 + H_0^2\,k_2$, and $\boldsymbol{H}' = \mathbf{O} \cdot \boldsymbol{H} = -H^1\,k_1 + H^2\,k_2$ that leads to the equalities $\tilde{m} = \tilde{m}'$ and $\boldsymbol{L}_{mag} = -\boldsymbol{L}_{mag}'$. Due to such rotation, angle $\phi$ between vector $\boldsymbol{H}_0'$ and fixed vertical axis $y$ shown in Figure 5, which is equal to $\beta$ for vector $\boldsymbol{H}_0$, becomes equal to $2\pi - \beta$, $0 \le \beta \le \pi/2$. With this in mind and considering the results of Section 3, we conclude that the detwinning process for $2\pi - \beta$ begins at the same $\tilde{H}_0^{cr}$ as is given in Table 2 for angle $\phi = \beta$, but, unlike the first case, it will be carried out in accordance with the second case discussed in Section 3. The magnetization curves will be exactly the same as for angle $\phi = \beta$.

Let us construct the average value of the projection of the magnetization on the axis along which the external magnetic field is directed:

$$\tilde{m} = <m_{||}> \frac{1}{S} \int_{\Omega^{(in)}} \left( \boldsymbol{m} \cdot \frac{\tilde{\boldsymbol{H}}_0}{\tilde{H}_0} \right) d\Omega^{(in)}, \tag{21}$$

where $S$ is the area of the considered domain.

Figure 7 demonstrates the dependencies of $\tilde{m}$ on the modulus $\tilde{H}_0$ for different directions (angles $\phi$) of the external magnetic field application, taking into account possible

detwinning. Five stages of the magnetization process are distinguished on each of the plotted curves. In the first stage, the 180-degree walls of the magnetic domains move proportionally to the applied magnetic field, and the magnetization depends linearly on this field. In the second stage, a jump in magnetization occurs due to a significant increase in the speed of the movement of these walls. At the third stage, these walls are annihilated in a critical field, the magnitude of which, $\tilde{H}_0$, is approximately 0.32 for an external field applied at angle $\phi = 0$, or 0.17 at angle $\phi = 45.84°$, or 0.13 at angle $\phi = 90°$ (see Figures 5–7 in [20]), and the magnetization vectors begin to turn gradually, tending to align with the applied external magnetic field. At the fourth stage, which is observed only for $\phi$ greater than $40°$, detwinning occurs, and the magnetization increases sharply again as a jump. For the curves presented in Figure 7, this takes place at the values of $\tilde{H}_0$ given in Table 2. At the last fifth stage, the magnetization reaches saturation. The magnetization curves obtained show essentially anisotropic magnetic properties in the twinned martensite of the $Ni_2MnGa$ alloy. Thus, we have solved the magnetic part of Problem 3, formulated in Section 4.2.

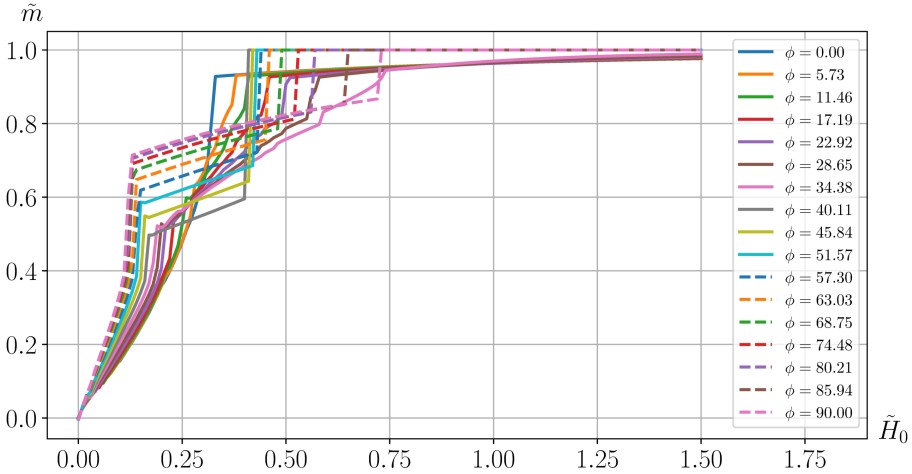

**Figure 7.** The magnetization curves for different directions of application of external magnetic field $\phi$ (in degrees), taking into account possible detwinning.

The detwinning process, which accompanies the magnetization process at certain values of angle $\phi$, leads to the occurrence of the structural deformation and can be realized as it has been shown, in accordance with two cases. Let us construct a deformed state for these cases. For detwinning process the strain tensor is given as $\mathbf{e} = (\mathbf{f}^{-T} \cdot \mathbf{f}^{-1} - \mathbf{g})/2$, where $\mathbf{f}^{-1} = \mathbf{f}_1^{-1}$ or $\mathbf{f}^{-1} = \mathbf{f}_2^{-1}$ is defined in basis $e_1$, $e_2$ by the relations (17) or (18). Using Expressions (17), this form can be concretized into an expression $\mathbf{e} = e^{11}\, e_1 e_1 + e^{22}\, e_2 e_2 + e^{12}\, (e_1 e_2 + e_2 e_1)$, where

$$e^{11} = (a^2 + c^2 + 2\,a)/2, \quad e^{22} = (a^2 + b^2 - 2\,a)/2, \quad e^{12} = [(1-a)\,c - (1+a)\,b]/2, \quad \text{the first case;}$$
$$e^{11} = (a^2 + b^2 - 2\,a)/2, \quad e^{22} = (a^2 + c^2 + 2\,a)/2, \quad e^{12} = [(1-a)\,c - (1+a)\,b]/2, \quad \text{the second case.}$$

By taking $a = 0.061582$, $b = 0.057899$, $c = 0.065498$ (see (12)), we obtain

$$e^{11} \approx 0.06, \quad e^{22} \approx -0.06, \quad e^{12} \approx 0.00 \quad \text{for the first case;}$$
$$e^{11} \approx -0.06, \quad e^{22} \approx 0.06, \quad e^{12} \approx 0.00 \quad \text{for the second case.} \tag{22}$$

These results correspond to the actual behavior of the sample. Indeed, for the detwinning process corresponding to the first case, element $2 - 3$ of twin $1 - 2 - 3$ in Figure 2 is converted to an element $2 - 3'$, which is an extension of element $1 - 2$. As a result, the sample increases its size in the direction of element $1 - 2 - 3'$ (in the direction of vector $e_1$) and decreases in the normal direction (in the direction of element $1' - 2 - 3$ or vector $e_2$). Therefore, the component $e^{11}$ of the strain tensor is positive, and $e^{22}$ is negative (see the first

line in (22)). For the detwinning process of the second case, element $1 - 2$ of twin $1 - 2 - 3$ in Figure 2 is converted to element $1' - 2$, which is an extension of element $2 - 3$. As a result, the sample increases in size in the direction of element $1' - 2 - 3$ (in the direction of vector $e_2$) and decreases in the normal direction (in the direction of element $1 - 2 - 3'$ or vector $e_1$). Therefore, the component $e^{22}$ of the strain tensor is positive, and $e^{11}$ is negative (see the second line in (22)), unlike the previous case. The values of these strains fully agree with the experimental data (see [33,34]) both in the first and second cases.

Since the detwinning in these two cases is realized with a simple shift, this process should take place without changing the volume. The obtained values of the strain tensors components fully correspond to this statement. In addition, it turned out that the vectors $e_1$, $e_2$, coinciding with both the axes of easy magnetization $c_1$, $c_2$ of the crystal and the axes of its anisotropy $p_1$, $p_2$, were the principal axes of the twinning and detwinning deformation processes.

It should be emphasized that deformation (22) occurs only if the conditions given in Table 2 are fulfilled: each angle $\phi$, which determines the direction of the external magnetic field application on the calculated area, corresponds to the intensity of this field. If this intensity is less than that given in Table 2, detwinning does not occur. With an account of the above-said, and based on the analysis performed immediately after Table 2 for angle $\phi$ varying from $90°$ to $360°$, we rewrite (22) as

$$e^{11} = 0.06 \, \Gamma(\phi, \tilde{H}_0), \quad e^{22} = -0.06 \, \Gamma(\phi, \tilde{H}_0), \quad e^{12} = 0.00,$$

$$\Gamma(\phi, \tilde{H}_0) = \begin{cases} H(\tilde{H}_0 - \tilde{H}_0^{cr}(\phi)) & \text{if} \quad 0 \le \phi \le \pi/2 \\ -H(\tilde{H}_0 - \tilde{H}_0^{cr}(\pi - \phi)) & \text{if} \quad \pi/2 \le \phi \le \pi \\ H(\tilde{H}_0 - \tilde{H}_0^{cr}(\phi - \pi)) & \text{if} \quad \pi \le \phi \le 3\pi/2 \\ -H(\tilde{H}_0 - \tilde{H}_0^{cr}(2\pi - \phi)) & \text{if} \quad 3\pi/2 \le \phi \le 2\pi \end{cases}, \tag{23}$$

where $\tilde{H}_0^{cr}(\zeta) = \infty$, if $0° \le \zeta \le 40°$ and $H(x)$ is the Heaviside's function, $H(x) = \begin{cases} 1 & \text{if} \quad x \ge 0 \\ 0 & \text{if} \quad x < 0 \end{cases}$. Having summarized the above, we can assert that the deformation part of Problem 3, formulated in Section 4.2, has been solved.

*5.2. A Polytwin Crystal*

In the previous subsection, we have considered a monotwin crystal under the action of the external magnetic field applied at different angles in the $xy$ plane. Now fixing the direction of the external magnetic field and placing the computational domain shown in Figure 3 at different angles to it, we will describe, using the curves in Figure 7, the magnetization of this representative composite region, which models a polycrystal with a different arrangement of twins in the plane, and deformed state of a polycrystal arising as a result of the detwinning processes in single crystals. This will be the solution to Problem 4 formulated in Section 4.2. With this approach, the commonly used structural models do not take into account the magnetic interaction, as well as the deformation interaction of these regions, and the question of these effects remains open.

In order to specify the location of single crystals in a polycrystal, we introduce, in addition to the orthonormal coordinate systems $(q_1, q_2)$ associated with single crystals, the general orthonormal coordinate system $(k_1, k_2)$ associated with the polycrystal (see Figure 8). The location of each single crystal in the polycrystal is determined by angle $\varphi$ between axis $q_2$ of this single crystal and vector $k_1$ of the coordinate system associated with the polycrystal common to all single crystals. The arcs between the black dots in Figure 8 span equal angles of $45°$.

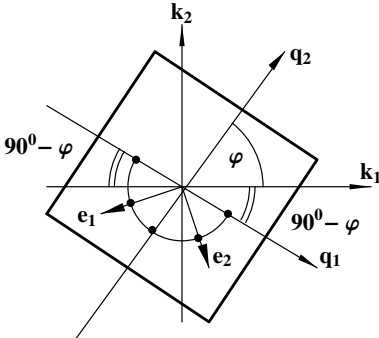

**Figure 8.** The position of a single crystal in a polycrystal; the orthonormal coordinate systems $(q_1, q_2)$ are associated with single crystals, the general orthonormal coordinate system $(k_1, k_2)$ are associated with the polycrystal.

To construct the magnetization curves for isotropic and anisotropic polycrystals, we use the following relation

$$m(\tilde{H}_0) = \left( \int_{\phi_1}^{\phi_2} \tilde{m}(\phi, \tilde{H}_0) \, d\phi \right) \Big/ \left( \int_{\phi_1}^{\phi_2} d\phi \right), \qquad (24)$$

where $\tilde{m}(\phi, \tilde{H}_0)$ is the values of magnetization that correspond to angle $\phi$, shown in Figure 3 for the single crystal, at point $\tilde{H}_0$. angle $\phi$ shown in Figure 3 is defined through angle $\varphi$ shown in Figure 8 using the relation $\phi = \varphi - \varphi_H$, where $\varphi$ is the angle between vector $k_1$ and vector $q_2$ and $\varphi_H$ is the angle between vector $H_0$ and vector $k_1$. Angle $\varphi$ defines the position of the single crystal in a polycrystal, and $\varphi_H$ defines the direction of the vector of external magnetic field $H_0$ with respect to the polycrystal. The relation (24) describes the effect of the magnetization of single crystals for the angles $\phi$ belonging to the segment $[\phi_1, \phi_2]$ on the magnetization of a polycrystal, which corresponds to the unit of angle $\phi$.

For an isotropic polycrystal relation (24) takes the following form

$$m(\tilde{H}_0) = \left( \int_{0}^{2\pi} \tilde{m}(\phi, \tilde{H}_0) \, d\phi \right) \Big/ \left( \int_{0}^{2\pi} d\phi \right). \qquad (25)$$

It is easy to see that this expression remains valid for any choice of $\varphi_H$. Let us choose any $\varphi_H$ from the closed segment $[0, 2\pi]$ and make it fixed. Assuming that $0 \le \varphi \le 2\pi$, we get:

$$\text{if} \quad \varphi_H \ge \varphi, \quad \text{then} \quad 0 \le \varphi \le \varphi_H \quad \text{and} \quad -\varphi_H \le \phi \le 0,$$
$$\text{if} \quad \varphi_H \le \varphi, \quad \text{then} \quad \varphi_H \le \varphi \le 2\pi \quad \text{and} \quad 0 \le \phi \le 2\pi - \varphi_H,$$
$$\text{and as a result} \quad -\varphi_H \le \phi \le 2\pi - \varphi_H.$$

In this case, the Expression (24) for an isotropic material is written as

$$m(\tilde{H}_0) = \left( \int_{-\varphi_H}^{2\pi - \varphi_H} \tilde{m}(\phi, \tilde{H}_0) \, d\phi \right) \Big/ \left( \int_{-\varphi_H}^{2\pi - \varphi_H} d\phi \right) \qquad (26)$$

and is a complete analog of equation (25). Since the established relation between (25) and (26) is valid for any $\varphi_H$, relation (25) does not depend on $\varphi_H$.

Let us go back to Expression (24). Dividing the segment $[\phi_1, \phi_2]$ into $n$ equal parts and supposing that $\tilde{m}$ is constant on each of the parts, we obtain from (24)

$$m(\tilde{H}_0) = \frac{1}{n} \sum_{j=1}^{n} \tilde{m}(\varphi_j - \varphi_H, \tilde{H}_0). \qquad (27)$$

We consider three types of polycrystalline samples: isotropic polycrystal, texture-oriented polycrystal—structure 1 and texture-oriented polycrystal—structure 2.

It is assumed that an isotropic polycrystal consists of 17 twinned single crystals of the same volume located at angles $\varphi_j = 0°, \ldots, 90°$ between axis $\boldsymbol{q}_2$ of the single crystals and vector $\boldsymbol{k}_1$ of the general coordinate system (see Figure 8). The magnetization curves corresponding to these 17 positions are shown in Figure 7 and are used to construct the magnetization curve for the isotropic polycrystal. Let us show that such a change in angle $\varphi$ is quite sufficient.

In connection with what has been said in the paragraphs following Table 2,

$$\tilde{m}(2\pi - \phi_j, \tilde{H}_0) = \tilde{m}(-\phi_j, \tilde{H}_0) = \tilde{m}(\phi_j, \tilde{H}_0), \quad \tilde{m}(\pi - \phi_j, \tilde{H}_0) = \tilde{m}(\phi_j, \tilde{H}_0),$$
$$\tilde{m}(\pi + \phi_j, \tilde{H}_0) = \tilde{m}(\phi_j, \tilde{H}_0), \quad \text{where} \quad \phi_j = \varphi_j - \varphi_H, \quad 0 \le \phi_j \le \pi/2. \tag{28}$$

We represent (27) for an isotropic material as

$$m(\tilde{H}_0) = \frac{1}{4n} \left[ \sum_{j=1}^{n} \tilde{m}(\phi_j, \tilde{H}_0) + \sum_{j=1}^{n} \tilde{m}(\pi - \phi_j, \tilde{H}_0) + \sum_{j=1}^{n} \tilde{m}(\pi + \phi_j, \tilde{H}_0) + \sum_{j=1}^{n} \tilde{m}(2\pi - \phi_j, \tilde{H}_0) \right],$$

where $\phi_j$ in each sum varies from $0°$ to $90°$. This expression takes into account a uniform distribution of the directions of the $\boldsymbol{q}_2$ vector of single crystals along a circle from $0°$ to $360°$, which must be performed for an isotropic polycrystal. Then, in accordance with (28), we have four identical sums in square brackets, and this expression eventually takes the following form for an isotropic material at value $n = 17$ given above:

$$m(\tilde{H}_0) = \frac{1}{17} \sum_{j=1}^{17} \tilde{m}(\phi_j, \tilde{H}_0), \quad \text{where} \quad 0 \le \phi_j \le \pi/2,$$

and is the simplest when $\varphi_H = 0$:

$$m(\tilde{H}_0) = \frac{1}{17} \sum_{j=1}^{17} \tilde{m}(\varphi_j, \tilde{H}_0). \tag{29}$$

Here $\tilde{m}(\varphi_j, \tilde{H}_0)$ is the value of magnetization at the point $\tilde{H}_0$ for the curve shown in Figure 7 that corresponds to angle $\phi_j$, at which the external magnetic field acts on the calculated domain for the single crystal. We use Expression (29) to construct the magnetization curve for the isotropic polycrystal shown in Figure 9.

There is a predominant direction of the martensitic structure orientation for textured polycrystals. It is assumed that structure 1 consists of 3 twinned single crystals, and structure 2 consists of 5 twinned single crystals of the same volume. These twinned crystals are located at angles $\varphi_j = 40.11°, 45.84°, 51.57°$ for structure 1 and $\varphi_j = 34.38°, 40.11°, 45.84°, 51.57°, 57.30°$ for structure 2 between vectors $\boldsymbol{q}_2$ of the twinned single crystals and $\boldsymbol{k}_1$ of the general coordinate system.

**Remark 2.** *This arrangement of twin crystals is not randomly chosen. The sets of single twin crystals differently located in space for structure 1 and structure 2 are grouped around the crystal whose angle $\varphi$ is about $45°$. This means that for this crystal, one element of the twin has axis $\boldsymbol{e}_1$ directed against vector $\boldsymbol{k}_1$, and the other element of the twin has axis $\boldsymbol{e}_2$ directed against vector $\boldsymbol{k}_2$ (see Figure 8). If the external magnetic field is applied along axis $\boldsymbol{k}_1$ and reaches the value of $\tilde{H}_0 \approx 0.4$ (see Table 2), this crystal shows a detwinning behavior and the element, which was directed along vector $-\boldsymbol{k}_2$, will be directed along vector $\boldsymbol{k}_1$. As a result, the magnetization in this crystal in the direction of vector $\boldsymbol{k}_1$ will increase in a jumpwise manner. In addition, as was shown above, the detwinning process causes in this crystal a structural strain, and in the direction of vector $\boldsymbol{k}_1$, its value is $e^{11} \approx 0.06$.*

*The values of $\tilde{H}_0 \approx 0.4$ and $e^{11} \approx 0.06$ are the control values of the magnetic and deformed behavior of structures 1 and 2 under the action of the external magnetic field in the direction of vector $\boldsymbol{k}_1$. Detwinning of other crystals in these structures under the action of the field in the same*

*direction causes the same elongation as in the above case but in a direction other than $k_1$. This, in accordance with (21), reduces the value of $\tilde{m}$ compared to the previous case, and the greater the difference of angle $\varphi_j$ from $45°$, the greater this decrease. As a result, the jump in the average value of magnetization determined by relation (24) should be less for structure 2 compared to structure 1 since structure 2 is structure 1 extended using the two elements, which are the most distant from the element defined by the angle of $45°$.*

The magnetization curve for the above anisotropic structures is described using the expression

$$m(\tilde{H}_0) = \frac{1}{n} \sum_{j=1}^{n} \tilde{m}(\varphi_j - \varphi_H, \tilde{H}_0),$$

where $n = 3$ for structure 1 and $n = 5$ for structure 2, $\varphi_H$ is the angle in the $xy$ plane between vectors $H_0$ and $k_1$, and $\tilde{m}(\varphi_j - \varphi_H, \tilde{H}_0)$ is the values of magnetization at the point $\tilde{H}_0$ for the curve shown in Figure 7 that correspond to angle $\varphi_j - \varphi_H$, at which the external magnetic field acts on the computational domain for the single crystal. By applying an external magnetic field along the $k_1$ vector ($\varphi_H = 0$) and in the direction of $45°$ to this axis ($\varphi_H = 45°$), we construct curves for an anisotropic polycrystalline material with structures 1 and 2 (see Figure 9).

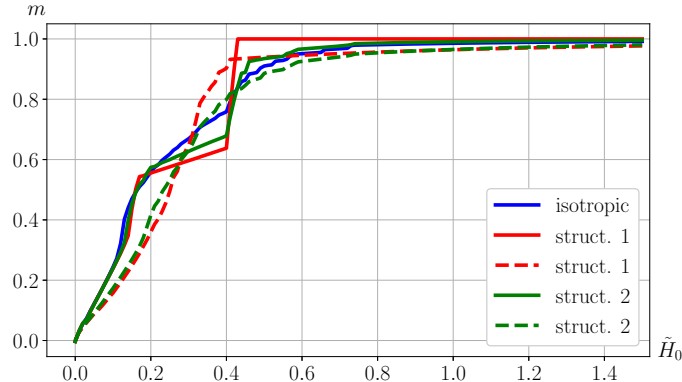

**Figure 9.** Magnetization curves for different polycrystals: isotropic and texture-oriented (structure 1 and structure 2). The solid lines show the magnetization curves for polycrystals when an external magnetic field is applied along the $k_1$ vector of these polycrystals. The dashed lines show the magnetization curves for textured polycrystals when a magnetic field is applied at an angle of $45°$ to vector $k_1$ of these polycrystals.

The solid lines in Figure 9 correspond to the magnetization curves for polycrystals when an external magnetic field is applied along the $k_1$ vector of these polycrystals. The dashed lines correspond to the magnetization curves for textured polycrystals when a magnetic field is applied at an angle of $45°$ to vector $k_1$ of these polycrystals (for an isotropic polycrystal, the magnetization curve completely coincides with the curve corresponding to the field acting along the $k_1$ vector).

When the external magnetic field is applied along vector $k_1$ (the solid lines), the curves for anisotropic structures 1 and 2 have jumps in the region $\tilde{H}_0 = 0.4$. Note that the jump for structure 1 is greater than the jump for structure 2. Such behavior is fully consistent with what was stated above in the *Remark* given before Figure 9. If, for structures 1 and 2, detwinning occurs simultaneously in all single crystals which form these structures, then in an isotropic material, detwinning occurs successively in single crystals with an increase of the external magnetic field since the intensity of this field is not enough for initiating simultaneous detwinning in all crystals. At a given time, detwinning occurs only in those crystals, the position of which and the relevant field strength satisfy the conditions

of Table 2. For this reason, the curve for an isotropic material in Figure 9 (solid blue line) consists of a series of small jumps of magnetization in the interval $0.4 \leq \tilde{H}_0 \leq 0.7$.

If an external magnetic field is applied at an angle of $45°$ to vector $\boldsymbol{k}_1$, then, as will be shown below, detwinning does not occur in structures 1 and 2, and there are no jumps on the curves in Figure 9 (dashed lines). The magnetization curves for these textured polycrystals differ depending on the direction of the applied magnetic field because such polycrystals are anisotropic.

Let us now define the deformed state that occurs in isotropic and anisotropic (structures 1 and 2) polycrystalline materials as the result of the detwinning process. As it has been stated at the end of the previous subsection, the deformations arising in the detwinning process are represented in basis $\boldsymbol{e}_1$, $\boldsymbol{e}_2$ as

$$\mathbf{e} = e^{11}\,\boldsymbol{e}_1\boldsymbol{e}_1 + e^{22}\,\boldsymbol{e}_2\boldsymbol{e}_2 + e^{12}\,(\boldsymbol{e}_1\boldsymbol{e}_2 + \boldsymbol{e}_2\boldsymbol{e}_1) \tag{30}$$

and have the following components (see (23))

$$e^{11} = 0.06\,\Gamma(\phi, \tilde{H}_0), \quad e^{22} = -0.06\,\Gamma(\phi, \tilde{H}_0), \quad e^{12} = 0.00. \tag{31}$$

In the basis of polycrystal $\boldsymbol{k}_1$, $\boldsymbol{k}_2$ (see Figure 8), the tensor (30) is represented as

$$\mathbf{e} = E^{11}\,\boldsymbol{k}_1\boldsymbol{k}_1 + E^{22}\,\boldsymbol{k}_2\boldsymbol{k}_2 + E^{12}\,(\boldsymbol{k}_1\boldsymbol{k}_2 + \boldsymbol{k}_2\boldsymbol{k}_1) \tag{32}$$

and from the equality of tensors in (30) and (32) we conclude that $E^{kp} = e^{ij}\,(\boldsymbol{e}_i \cdot \boldsymbol{k}_k)(\boldsymbol{e}_j \cdot \boldsymbol{k}_p)$ or, given that $e^{12} = 0$ for any case of detwinning process (see (31)),

$$\begin{aligned}
E^{11} &= e^{11}\,(\boldsymbol{e}_1 \cdot \boldsymbol{k}_1)(\boldsymbol{e}_1 \cdot \boldsymbol{k}_1) + e^{22}\,(\boldsymbol{e}_2 \cdot \boldsymbol{k}_1)(\boldsymbol{e}_2 \cdot \boldsymbol{k}_1), \\
E^{22} &= e^{11}\,(\boldsymbol{e}_1 \cdot \boldsymbol{k}_2)(\boldsymbol{e}_1 \cdot \boldsymbol{k}_2) + e^{22}\,(\boldsymbol{e}_2 \cdot \boldsymbol{k}_2)(\boldsymbol{e}_2 \cdot \boldsymbol{k}_2), \\
E^{12} &= e^{11}\,(\boldsymbol{e}_1 \cdot \boldsymbol{k}_1)(\boldsymbol{e}_1 \cdot \boldsymbol{k}_2) + e^{22}\,(\boldsymbol{e}_2 \cdot \boldsymbol{k}_1)(\boldsymbol{e}_2 \cdot \boldsymbol{k}_2).
\end{aligned} \tag{33}$$

From Figure 8 we have

$$\begin{aligned}
\boldsymbol{e}_1 \cdot \boldsymbol{k}_1 &= -\cos(\varphi - 45°), \quad \boldsymbol{e}_1 \cdot \boldsymbol{k}_2 = -\sin(\varphi - 45°), \\
\boldsymbol{e}_2 \cdot \boldsymbol{k}_1 &= \sin(\varphi - 45°), \quad \boldsymbol{e}_2 \cdot \boldsymbol{k}_2 = -\cos(\varphi - 45°)
\end{aligned}$$

and then expressions in (33) take the following form

$$E^{11} = e^{11}\,\cos^2(\varphi - 45°) + e^{22}\,\sin^2(\varphi - 45°), \quad E^{22} = e^{11}\,\sin^2(\varphi - 45°) + e^{22}\,\cos^2(\varphi - 45°),$$
$$E^{12} = e^{11}\,\sin(\varphi - 45°)\,\cos(\varphi - 45°) - e^{22}\,\sin(\varphi - 45°)\,\cos(\varphi - 45°)$$

or, allowing for that $\cos^2\alpha = [1 + \cos(2\,\alpha)]/2$, $\sin^2\alpha = [1 - \cos(2\,\alpha)]/2$, $\sin\alpha\,\cos\alpha = (1/2)\sin(2\,\alpha)$,

$$E^{11} = \frac{1}{2}\,(e^{11} + e^{22}) + \frac{1}{2}\,(e^{11} - e^{22})\,\sin(2\,\varphi), \quad E^{22} = \frac{1}{2}\,(e^{11} + e^{22}) - \frac{1}{2}\,(e^{11} - e^{22})\,\sin(2\,\varphi),$$
$$E^{12} = -\frac{1}{2}\,(e^{11} - e^{22})\,\cos(2\,\varphi)$$

and considering (31), we obtain

$$E^{11}(\phi, \varphi, \tilde{H}_0) = 0.06\,\Gamma(\phi, \tilde{H}_0)\,\sin(2\,\varphi), \quad E^{22}(\phi, \varphi, \tilde{H}_0) = -0.06\,\Gamma(\phi, \tilde{H}_0)\,\sin(2\,\varphi),$$
$$E^{12}(\phi, \varphi, \tilde{H}_0) = -0.06\,\Gamma(\phi, \tilde{H}_0)\,\cos(2\,\varphi). \tag{34}$$

As explained earlier, $\phi = \varphi - \varphi_H$ (see the explanation after the Relation (24)).

The Relations (34) allow us to construct the deformed state that occurs in an isotropic or anisotropic polycrystal during detwinning in single crystals. To do this, we use a relation similar to (24),

$$\hat{E}^{ij}(\varphi_H) = \left( \int_{\varphi_1}^{\varphi_2} E^{ij}(\phi, \varphi, \tilde{H}_0) \, d\varphi \right) \bigg/ \left( \int_{\varphi_1}^{\varphi_2} d\varphi \right). \tag{35}$$

Dividing the segment $[\varphi_1, \varphi_2]$ into $n$ equal parts, $\Delta\varphi = (\varphi_2 - \varphi_1)/n$, supposing that $\varphi_j = \varphi_1 + (j + 0.5)\,\Delta\varphi$, calculating $E^{ij}$ at these points and assuming that it is constant on each such part, we obtain

$$\hat{E}^{ij}(\varphi_H) = \frac{1}{n} \sum_{j=0}^{n-1} E^{ij}(\varphi_j - \varphi_H, \varphi_j, \tilde{H}_0). \tag{36}$$

The Expression (36) is found to be more suitable for an anisotropic material.

For the isotropic material, we use the relation (35). Consider the cases when the external magnetic field of strength $\tilde{H}_0 = 0.46$ and $\tilde{H}_0 = 1.0$ is applied at the angles of $\varphi_H = 0°$ and $\varphi_H > 0°$. Then, for $\varphi_H = 0°$, $\phi = \varphi - \varphi_H = \varphi$ and in accordance with (23) and the Table 2 we have for $\tilde{H}_0 = 0.46$

$$\Gamma(\varphi) = \begin{cases} 0 & \text{if} \quad 0 \le \varphi \le \alpha_1 \\ 1 & \text{if} \quad \alpha_1 \le \varphi \le \alpha_2 \\ 0 & \text{if} \quad \alpha_2 \le \varphi \le \pi/2 \end{cases}, \quad \Gamma(\varphi) = -\begin{cases} 0 & \text{if} \quad \pi/2 \le \varphi \le \pi - \alpha_2 \\ 1 & \text{if} \quad \pi - \alpha_2 \le \varphi \le \pi - \alpha_1 \\ 0 & \text{if} \quad \pi - \alpha_1 \le \varphi \le \pi \end{cases},$$

$$\Gamma(\varphi) = \begin{cases} 0 & \text{if} \quad \pi \le \varphi \le \pi + \alpha_1 \\ 1 & \text{if} \quad \pi + \alpha_1 \le \varphi \le \pi + \alpha_2 \\ 0 & \text{if} \quad \pi + \alpha_2 \le \varphi \le 3\pi/2 \end{cases}, \quad \Gamma(\varphi) = -\begin{cases} 0 & \text{if} \quad 3\pi/2 \le \varphi \le 2\pi - \alpha_2 \\ 1 & \text{if} \quad 2\pi - \alpha_2 \le \varphi \le 2\pi - \alpha_1 \\ 0 & \text{if} \quad 2\pi - \alpha_1 \le \varphi \le 2\pi \end{cases},$$

where, as it follows from Table 2, $\alpha_1 = 40°$, $\alpha_2 = 63.03°$. With this in mind and substituting (34) into (35) we get

$$\hat{E}^{11} = \frac{0.06}{2\pi} \left( \int_{\alpha_1}^{\alpha_2} \sin(2\varphi) \, d\varphi - \int_{\pi-\alpha_2}^{\pi-\alpha_1} \sin(2\varphi) \, d\varphi + \right.$$
$$\left. + \int_{\pi+\alpha_1}^{\pi+\alpha_2} \sin(2\varphi) \, d\varphi - \int_{2\pi-\alpha_2}^{2\pi-\alpha_1} \sin(2\varphi) \, d\varphi \right). \tag{37}$$

It is easy to show that each of the last three integrals with their signs is equal to the first integral. As a result,

$$\hat{E}^{11} = 2 \frac{0.06}{\pi} \int_{\alpha_1}^{\alpha_2} \sin(2\varphi) \, d\varphi = \frac{0.06}{\pi} \left[ \cos(2\alpha_1) - \cos(2\alpha_2) \right]. \tag{38}$$

Substituting the above values of angles in this expression, we find that at $\varphi_H = 0°$ and $\tilde{H}_0 = 0.46$, $\hat{E}^{11} = 0.015$ and, as it follows from (34), $\hat{E}^{22} = -0.015$. It also follows from (34) that $\hat{E}^{12}$ is represented in the case under consideration as (37), in which the sine is replaced by cosine and the minus sign is placed in front of the whole expression. It can be easily shown that the terms in the obtained expression are mutually eliminated at any $\alpha_1$ and $\alpha_2$ and, as a result, $\hat{E}^{12} = 0$.

Now, if $\tilde{H}_0 = 1.0$, then, in accordance with Table 2, we have for $\varphi_H = 0°$ that $\alpha_2 = \pi/2$ in the Expression (37) for $\hat{E}^{11}$ and as a result, $\hat{E}^{11} = 0.022$, $\hat{E}^{22} = -0.022$, $\hat{E}^{12} = 0$. The increase in the modulus of these components with increasing the external magnetic field is explained by the involvement of additional regions in the detwinning process (see Table 2). The expression for $\Gamma(\varphi)$, which is not quite correct in this case, can be readily converted, but due to the use of Relation (37), there is no need to do this.

For $\varphi_H > 0°$, $\phi = \varphi - \varphi_H$ and in accordance with (23) and the Table 2 we have for $\tilde{H}_0 = 0.46$ that

$$
\Gamma(\varphi) = \begin{cases}
1 & \text{if} \quad \varphi_H + \alpha_1 \leq \varphi \leq \varphi_H + \alpha_2 \\
-1 & \text{if} \quad \pi + \varphi_H - \alpha_2 \leq \varphi \leq \pi + \varphi_H - \alpha_1 \\
1 & \text{if} \quad \pi + \varphi_H + \alpha_1 \leq \pi + \varphi_H + \alpha_2 \\
-1 & \text{if} \quad \pi + \varphi_H - \alpha_2 \leq \pi + \varphi_H - \alpha_1
\end{cases} \quad ,
$$

and $\Gamma(\varphi) = 0$ for all other $\varphi$. Then, equation (35), in view of Relations (34), is written for $\hat{E}^{11}$ as

$$
\hat{E}^{11} = \frac{0.06}{2\pi} \left( \int_{\varphi_H + \alpha_1}^{\varphi_H + \alpha_2} \sin(2\varphi)\, d\varphi - \int_{\pi + \varphi_H - \alpha_2}^{\pi + \varphi_H - \alpha_1} \sin(2\varphi)\, d\varphi + \right.
$$
$$
\left. \int_{\pi + \varphi_H + \alpha_1}^{\pi + \varphi_H + \alpha_2} \sin(2\varphi)\, d\varphi - \int_{2\pi + \varphi_H - \alpha_2}^{2\pi + \varphi_H - \alpha_1} \sin(2\varphi)\, d\varphi \right). \quad (39)
$$

It is easy to show that this equality can be represented as

$$
\hat{E}^{11} = \frac{0.06}{\pi} \left( \int_{\alpha_1}^{\alpha_2} \sin(2\varphi + 2\varphi_H)\, d\varphi + \int_{\alpha_1}^{\alpha_2} \sin(2\varphi - 2\varphi_H)\, d\varphi \right),
$$

or as

$$
\hat{E}^{11} = \frac{0.06}{\pi} \int_{\alpha_1}^{\alpha_2} [\sin(2\varphi + 2\varphi_H) + \sin(2\varphi - 2\varphi_H)]\, d\varphi,
$$

or given at last that $\sin A + \sin B = 2 \sin \dfrac{A+B}{2} \cos \dfrac{A-B}{2}$, as

$$
\hat{E}^{11}(\varphi_H) = 2\, \frac{0.06}{\pi} \cos(2\varphi_H) \int_{\alpha_1}^{\alpha_2} \sin(2\varphi)\, d\varphi. \quad (40)
$$

As it follows from (34), $\hat{E}^{22} = -\hat{E}^{11}$ and $\hat{E}^{12}$ is represented in the case under consideration as (39) with the replacement of the sine by the cosine and the addition of the minus sign in front of the whole expression

$$
\hat{E}^{12} = -\frac{0.06}{2\pi} \left( \int_{\varphi_H + \alpha_1}^{\varphi_H + \alpha_2} \cos(2\varphi)\, d\varphi - \int_{\pi + \varphi_H - \alpha_2}^{\pi + \varphi_H - \alpha_1} \cos(2\varphi)\, d\varphi + \right.
$$
$$
\left. \int_{\pi + \varphi_H + \alpha_1}^{\pi + \varphi_H + \alpha_2} \cos(2\varphi)\, d\varphi - \int_{2\pi + \varphi_H - \alpha_2}^{2\pi + \varphi_H - \alpha_1} \cos(2\varphi)\, d\varphi \right).
$$

This equality is reduced to

$$
\hat{E}^{12} = \frac{0.06}{\pi} \int_{\alpha_1}^{\alpha_2} [\cos(2\varphi + 2\varphi_H) - \cos(2\varphi - 2\varphi_H)]\, d\varphi,
$$

or, given that $\cos A - \cos B = 2 \sin \dfrac{A+B}{2} \sin \dfrac{B-A}{2}$, to

$$
\hat{E}^{12}(\varphi_H) = 2\, \frac{0.06}{\pi} \sin(2\varphi_H) \int_{\alpha_1}^{\alpha_2} \sin(2\varphi)\, d\varphi. \quad (41)
$$

Relations (40) and (41) are the general expressions that define the components of the strain tensor for an isotropic shape memory material during detwinning. The magnitudes of these components depend both on the magnitude of the applied external magnetic field, which in accordance with Table 2 determines the value of angle $\alpha_2$, and on the direction of the field application (on angle $\varphi_H$). As it follows from Table 2, $\alpha_2 = 63.03°$ when $\tilde{H}_0 = 0.46$ and $\alpha_2 = 90°$ when $\tilde{H}_0 = 1.00$. At the same time, angle $\alpha_1$ remains unchanged, $\alpha_1 = 40°$. When $\varphi_H = 0$ Expression (40) is similar to (38) and from the Expression (41) it

follows that $\hat{E}^{12} = 0$ as was noted earlier. Although the components of the strain tensor in the orthonormal basis $k_i$ (see Figure 8) depend on angle $\varphi_H$ between vectors $k_1$ and $H_0$, in the orthonormal basis $\gamma_i$, in which vector $\gamma_1$ is directed along vector $H_0$, these components remain unchanged. This can be easily shown by analogy with establishing the relationship between $e^{ij}$ and $E^{kp}$ in the Expressions (30) and (32). Therefore, such material is called isotropic.

Let us now consider the deformed state of an anisotropic material with a shape memory effect during its detwinning. We assume that twinned single crystals of the same volume are continuously distributed in a polycrystal at $\varphi_1 \leq \varphi \leq \varphi_2$, and the external magnetic field is applied at angle $\varphi_H$. We will consider two special cases of textured polycrystals for which the magnetization curves have been already constructed: structure 1 for $\varphi_1 = 40.11°$ and $\varphi_2 = 51.57°$, and structure 2 for $\varphi_1 = 34.38°$ and $\varphi_2 = 57.30°$. For these two cases, we present an algorithm for determining the deformed state during the dissipation of twins (during detwinning) for the two angles $\varphi_H$ of the application of an external magnetic field: $\varphi_H = 0$ and $\varphi_H = 45°$, and two $\tilde{H}_0$: $\tilde{H}_0 = 0.46$ and $\tilde{H}_0 = 1.00$. The deformed state during detwinning for any other case of anisotropy and the direction of action of the external magnetic field can be determined using this algorithm.

Let us analyze the deformation behavior of structure 1 at $\varphi_H = 0$, $\tilde{H}_0 = 0.46$ and summarize the obtained findings. (1) Since $\phi = \varphi - \varphi_H$ then $\phi = \varphi$ and $\varphi_1 \leq \phi \leq \varphi_2$ where $\varphi_1 = 40.11°$ and $\varphi_2 = 51.57°$. (2) Such a change in $\phi$ corresponds to the first line in (23). As a result, we have $\Gamma(\phi, \tilde{H}_0) = H(0.46 - \tilde{H}_0^{cr}(\phi))$. (3) As it follows from Table 2, $\Gamma = 1$ for $\tilde{H}_0 = 0.46$ if $\phi = \varphi \in U$ where the set $U = [40°, 63.03°]$. Since for structure 1 $\varphi \in U_1$ where the set $U_1 = [40.11°, 51.57°]$ then it follows from the intersection of the sets $U$ and $U_1$, $U \cap U_1$, that $\varphi_1 = 40.11°$ and $\varphi_2 = 51.57°$. (4) As a result, we find from the Expressions (34) and (35) that

$$\hat{E}^{11} = a\,S, \quad \hat{E}^{22} = -a\,S, \quad \hat{E}^{12} = -a\,C,$$

$$a = 0.06/(\varphi_2 - \varphi_1), \quad S = \int_{\varphi_1}^{\varphi_2} \sin(2\,\varphi)\,d\varphi, \quad C = \int_{\varphi_1}^{\varphi_2} \cos(2\,\varphi)\,d\varphi \tag{42}$$

and for structure 1, $\varphi_H = 0$, $\tilde{H}_0 = 0.46$ the detwinning strain tensor has the following components in the basis $k_i$ (see Figure 8): $\hat{E}^{11} = 0.06$, $\hat{E} = -0.06$, $\hat{E}^{12} = 0.0017$.

Now, let us analyze the deformation behavior of structure 1 at $\varphi_H = 0$, $\tilde{H}_0 = 1.00$. Everything that is stated in item (1) of the previous case remains valid. In accordance with item (2) $\Gamma(\phi, \tilde{H}_0) = H(1.00 - \tilde{H}_0^{cr}(\phi))$ and with item (3) the set $U$ for $\tilde{H}_0 = 1.00$ has the form $U = [40°, 90°]$. Since the set $U_1$ remains unchanged, then the set, which is the intersection of sets $U$ and $U_1$, remains unchanged, too. As a result, the angles $\varphi_1$ and $\varphi_2$ are the same as in the previous case, and with reference to item (4), we have the same components of the detwinning strain tensor as in the above case.

If the external magnetic field acts on structure 1 at angle $\varphi_H = 45°$, then $\phi = \varphi - \varphi_H$ will change, considering that $\varphi \in U_1$ where the set $U_1 = [40.11°, 51.57°]$, in the interval $\phi \in [-4.89°, 6.57°]$ which corresponds to the first and fourth areas of change in $\phi$ in the Expression (23): $\phi \in [0°, 6.57°]$ and $\phi \in [355.11°, 360°]$. Since $\tilde{H}_0^{cr}(\zeta) = \infty$, if $0° \leq \zeta \leq 40°$, where $\zeta = \phi$ in the first area and $\zeta = 2\pi - \phi$ in the second area, then $\Gamma = 0$ in any of these cases and detwinning does not occur at any magnitude of the external magnetic field.

All of the above concerning the deformation behavior during the detwinning of structure 1 at $\varphi_H = 0$ and $\tilde{H}_0 = 0.46$ remains valid for structure 2 involving the replacement of the set $U_1$ by the set $U_2$, where the set $U_2 = [34.38°, 57.30°]$ is the set that describes structure 2. The consequence of the intersection of the sets $U$ and $U_2$ (instead of $U_1$) are the values of the angles $\varphi_1 = 40°$ and $\varphi_2 = 57.30°$, for which we eventually determine from the Expressions (42) that $\hat{E}^{11} = 0.0586$, $\hat{E} = -0.0586$, $\hat{E}^{12} = 0.0075$.

By performing analysis similar to that made for structure 1 at $\varphi_H = 0$ and $\tilde{H}_0 = 1.00$, and also at $\varphi_H = 45°$, we obtain similar results for structure 2: at $\varphi_H = 0$ and $\tilde{H}_0 = 1.00$,

the deformed state remains exactly the same as at $\varphi_H = 0$ and $\tilde{H}_0 = 0.46$, and at $\varphi_H = 45°$, and detwinning does not occur.

The results of the performed deformation analysis are presented in Table 3. Here $E^{11}$ is the average tensile strain of the considered structure in the direction of vector $\boldsymbol{k}_1$ (see Figure 8) caused by the detwinning process (for an isotropic material, this vector can have any direction). As it follows from this table, the detwinning of an isotropic material occurs sequentially and is determined by an increase in the strength of the applied external magnetic field. In structures 1 and 2, complete detwinning occurs already at $\tilde{H}_0 = 0.46$, when the field is applied along vector $\boldsymbol{k}_1$, and the value of $E^{11}$ for the first structure is greater than that of the second one. When the field is applied at the angle of $45°$ to vector $\boldsymbol{k}_1$, the detwinning does not occur at any external field strength.

**Table 3.** The results of the performed deformation analysis

| | $\boldsymbol{\varphi_H = 0}$ | | $\boldsymbol{\varphi_H = 45°}$ | |
|---|---|---|---|---|
| | $\tilde{H}_0 = 0.46$ | $\tilde{H}_0 = 1.0$ | $\tilde{H}_0 = 0.46$ | $\tilde{H}_0 = 1.0$ |
| Isotrop., $E^{11}$ | 0.015 | 0.022 | 0.015 | 0.022 |
| Structure 1, $E^{11}$ | 0.06 | 0.06 | 0 | 0 |
| Structure 2, $E^{11}$ | 0.0586 | 0.0586 | 0 | 0 |

The results of the deformation analysis are consistent with the explanations given above in the *Remark* and when discussing the curves in Figure 9.

## 6. Conclusions

In this article, within the framework of the theory of micromagnetism, the problem of magnetization of a single twinned martensitic crystal of the shape memory $Ni_2MnGa$ alloy using the finite element method was solved. The variational equations were put in accordance with the Landau–Lifshitz–Gilbert equation and other differential equations and boundary conditions of the theory of micromagnetism. This made it possible to reduce the requirements for the smoothness of the problem solution. Magnetization curves were plotted for different angles of application of the magnetic field to the anisotropy axes of the twin variants.

A condition for detwinning of a shape memory ferromagnetic alloy was proposed for the case when only a magnetic field acts on the body. This simplest way to simulate the detwinning process is based on the calculation of the mass magnetic moment. When this moment reaches each element of the twin, a critical value, determined experimentally, detwinning occurs. It is shown that this critical moment corresponds to different strengths of external magnetic field $\tilde{H}_0^{cr}$ depending on the direction of its action with respect to the twin (from angle $\phi$) and quantitative compliance has been established between $\tilde{H}_0^{cr}$ and $\phi$. The disappearance of the twin (detwinning) causes significant structural deformations in the material, which affects the magnetization curves of a shape memory ferromagnetic alloy in addition to the mechanisms that determine the magnetization of conventional ferromagnetic alloys.

The curves constructed on the basis of the proposed model showed that the change in the main mechanisms of magnetization, such as the movement and interaction of 180-degree magnetic domain walls, the rotation of the local magnetization vectors, and the occurrence of the structural deformation associated with the detwinning process, led to kinks in the magnetization curves, which occur at different values of the external magnetic field depending on the direction of this magnetic field.

Based on the magnetization curves obtained for the single-crystal and deformation state, which corresponds to the detwinning state of such structure, the deformed states and the magnetization curves were constructed for various types of polycrystals (isotropic and textured), which are the structures formed of single crystals. For texture-oriented

polycrystals, the magnetization curves and deformation states differed depending on the direction of the applied magnetic field because such polycrystals are anisotropic.

**Author Contributions:** A.A.R.: conceptualization, investigation, formal analysis, writing—original draft preparation; O.S.S.: conceptualization, investigation, formal analysis, writing—original draft preparation. All authors have read and agreed to the published version of the manuscript.

**Funding:** This work was fulfilled under financial support of the Russian Foundation for Basic Research through Grant 20-01-00031. The authors express their sincere thanks for this support.

**Institutional Review Board Statement:** Not applicable.

**Informed Consent Statement:** Not applicable.

**Data Availability Statement:** Not applicable.

**Acknowledgments:** The study was carried out using the ICMM UB RAS supercomputer Triton.

**Conflicts of Interest:** The authors declare no conflict of interest.

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
