# Peer review of "Microstructural Model of Magnetic and Deformation Behavior of Single Crystals and Polycrystals of Ferromagnetic Shape Memory Alloy"

_magnetochemistry, doi:10.3390/magnetochemistry9020040_

Round 1

Reviewer 1 Report

This is a numerical study on the magnetization of a single twinned martensitic crystal of the Ni2MnGa alloy with shape memory using the finite element method. This work may be published after the authors address a few concerns:

1.       The LLG formulation can sometimes include a term for the thermal fluctuation of the magnetic moments. What was the authors’ rationale for excluding thermal effects from their analysis?

2.       The authors do not explicitly state which numerical technique was used for time stepping. Was it the forward Euler method? The backward Euler method? Or perhaps something else? This information needs to be included in the manuscript. Additionally, the authors mention the total number of finite elements used, however, it would also be useful to include information regarding the mesh size.

3.       The abstract needs to be re-written to give a brief summary of the results. The abstract, as presently constructed, only describes the methodology used by the authors.

4.       The conclusions section does not include any conclusions. Please specify what you learned from analyzing the results. What was the “proposed condition”? What did you learn from the magnetization curves? What did the deformation analysis tell you?

5.       How do these results relate to the wider literature? How can the information gleaned by the authors be useful elsewhere? Is there any experimental data that the authors can compare their results to? Or can the authors use their model to explain some experimental observations?

Author Response

We thank the reviewer for work, comments and suggestions. We took them into account in the revised version of the article and believe that this will improve its perception.

Reviewer 2 Report

The novelty of this work needs to be stressed and there are some issues that should be clarified in the present version of the manuscript.

1. Why do the magnetization vectors only rotate in a ferromagnetic alloy with shape memory? What role does the shape memory play during the detwinning process?

2. To compare to the experimental results, the authors referred to the specified results in Ni51Mn29Ga20 sample. However, the atom ratio changes a lot from the full Heusler alloy Ni2MnGa. Is Ni51Mn29Ga20 still a Heusler alloy with a shape memory? And why do the authors mainly focus on the Ni2MnGa material system? How about other Heusler alloy systems? Are the calculation results still applicable? 

3. On P.10, the authors define the domain wall thickness as sqrt(A_exch/K_anis). Generally, this value is defined as the magnetocrystalline exchange length, which is used as a reference for determining the size of the meshes for the related calculation. As for the domain wall thickness, it is usually larger than the exchange length. Please check “Adv. Phys., 54, 585-713, 2005” for reference. The DW thickness should be confirmed by considering the balance between the exchange energy and anisotropy energy. How do the authors consider the definition of the domain wall thickness? Is the value used in the manuscript accurate?

4. The recent progress on the related research and the significance of this work should be further stressed. 

Author Response

(The authors gave the same response as above.)

Round 2

Reviewer 1 Report

The authors have addressed my concerns. I now recommend that the manuscript be published.